# Mitigating Water Shortage via Hydrological Modeling in Old and New Cultivated Lands West of the Nile in Egypt

Abir M. Badr [1,*], Fadi Abdelradi [2], Abdelazim Negm [3,*] and Elsayed M. Ramadan [3]

1 General Administration of Irrigation of West Sharkia, Ministry of Water Resources and Irrigation, Zagazig 44519, Egypt
2 Department of Agricultural Economics, Faculty of Agriculture, Cairo University, Giza 12613, Egypt; fadi.abdelradi@agr.cu.edu.eg
3 Water and Water Structures Engineering Department, Faculty of Engineering, Zagazig University, Zagazig 44519, Egypt; smokhtar@zu.edu.eg
* Correspondence: a.badr021@eng.zu.edu.eg (A.M.B.); amnegm@zu.edu.eg (A.N.)

**Abstract:** Middle East and North Africa (MENA) regions are increasingly concerned about water scarcity. Egypt, one of the arid MENA nations that relies primarily on Nile water, faces a water scarcity issue because of a mismatch between demand and supply. This study presents an integrated executive system for managing water resources in two regions of Egypt that have traits with many MENA regions facing water scarcities. Hydrological modeling is required for the modeling of water resources, and model calibration procedures should be implemented to compare the simulated values to the observed and measured values to minimize model errors. The Water Evaluation and Planning (WEAP) model was used in this study to simulate the network systems of Egypt's Minia Governorate on the western bank of the Nile's narrow valley and Nubariya in the West Nile Delta, the lower reaches of the Nile. Using field data and experience, as well as other inputs, geographic information system (GIS) software digitized streams using satellite-interpreted data. The models were run, calibrated, and validated. The main calibration objective was to reduce the discrepancy between the actual and modeled flows as much as possible. Nash–Sutcliffe efficiency (NSE), percentage BIAS (PBIAS), volumetric efficiency (VE), and agreement index (d) values were calculated for three calibration cases. For anticipating water shortages until 2050, two scenarios were examined: (1) climate change scenarios based on historical climatic data from 1960 to 1990 and from 1991 to 2020, which led to a prediction scenario (2021–2050) of increasing temperature in the areas leading to evapotranspiration (ET) increases of 5.42% and 5.13% and (2) canal lining scenarios, which found a flow saving in the areas, showing that we can overcome the anticipated water shortage progress if canal lengths are rehabilitated by 10% and 25% in Minia and Nubariya.

**Keywords:** sustainability; water management; WEAP; calibration; validation and scenarios



## 1. Introduction

Water shortage is one of the major issues many MENA countries are currently dealing with. Due to the unequal distribution of water throughout the world, sustainability and efficient water use are crucial, and they are especially crucial for nations facing water scarcity. Sustainable water resource management and providing sufficient safe, clean water to all people are the targets of sustainable development [1]. Egypt is a typical case, facing a rapidly growing population [2] and several advancements in many sectors, such as establishing the new capital. Egypt is the nation furthest downstream in the Nile Basin, and most of its freshwater inflow (about 97%) comes from outside its boundaries [3,4]. According to the Egyptian national plan 2037–2050, for various sectors in Egypt, the yearly water shortage is about 21 billion cubic meters (BCM), and is continuously growing. At the same time, the per capita annual water availability will be only 500 cubic meters by 2025, which infers absolute water scarcity [4–7]. All available water resources are used

to meet all sectors' demands. To meet the needs of all sectors, all water resources are utilized. Therefore, Egypt's current national plan reflects a transition from conventional approaches to demand management. The control of water resources has also moved from being conducted locally to being conducted nationally. The sustainability of water resource systems will necessitate considering the detrimental effects of population increase, urbanization, and national and local objectives [8].

A modern-world requirement due to the rapid rise in climate fluctuations and water demand is the sustainable management of water resources through optimal water distribution [9]. Adapting to the expected stage of rapid change brought on by increasing urbanization and economic, physical, and social advancements is a concern for the future. In addition to climate change uncertainties and the transboundary nature of the hydrological regime, the worry is that these anticipated rapid developments would put strain on water resources. Given this, in order to forecast the hydrological behavior of surface and subsurface systems in response to human activity and climate change, a vast variety of models have been constructed [10].

WEAP is a hydrological model developed by the Stockholm Environment Institute. It is a data-processing tool that enables water managers to evaluate different management scenarios in an area by simulating a wide range of changes [11–14]. This tool in water resources management includes a spatial geodatabase in a geographic information system (GIS). Although it does not have any capacity for calculating separate water balance [15], numerous administrators and scholars throughout the world have utilized WEAP [16–18]; there are wide-ranging studies that have used the WEAP program [1,19–22].

The number of nexus studies that have employed the WEAP program is quite varied. In order to understand the Water–Energy–Food–Environment (WEFE) nexus for climate change adaptation in the Urmia Lake Basin in northwest Iran, Nasrollahi et al. combined LEAP and WEAP models and investigated potential future climate scenarios [22]. Simons et al. used WEAP's water quality modeling capabilities to analyze water reuse processes in the Segura River Basin in Spain and Europe using a virtual tracer technique [23]. Alamanos et al. combined hydro-economic and water quality modeling in the Lake Karla watershed in central Greece to investigate the possibility of improving water quality through various irrigation and water conservation measures/policies [21]. Psomas et al. and Maliehe et al. integrated models using WEAP and SWAT software. For the upper Pinios river basin and the towns of Trikala and Karditsa in Thessaly, Greece, Psomas combined WEAP, SWAT, and GIS [19]. In addition, for the Lesotho region of the South Phuthiatsana watershed, Maliehe et al. integrated SWAT and WEAP [20].

To analyze the dynamic between quantity and quality in the Vieja river basin, Colombia, Jaramillo et al. combined the QUAL2K and WEAP models [24]. Ramadan et al. assessed the water deficit in the Egyptian province of Sharkia using WEAP [25]. WEAP applications support many case studies all over the world and can be found on the website www.weap21.org (accessed on 27 May 2023).

In a river system, the WEAP model successively establishes a mass flow balance while allowing inflows and abstractions [26]. The priority assigned to each demand location ranges from 1 to 99, with 1 denoting the highest priority and 99 the lowest. When there is a water shortage, this algorithm is designed to gradually reduce the amount of water allocated to the demand sites that receive the lowest priority. More details of the model are available in [12,16,17].

In this study, two WEAP models were integrated with a GIS. Mathematical verifications and analytical prediction of climate change were assessed for their potential to inform policies to alleviate the seasonal water deficit in old lands, i.e., Minia (West Nile Valley) and new lands, i.e., Nubariya (West Nile Delta), two separate regions of Egypt. The findings of this study are intended to shed light on potential uses of hydrological modeling and WEAP models, which will aid in developing and planning evaluation procedures globally and in locations where water resources are already heavily stressed. In other words, it will provide methods for better water resource management [27].

## 2. Identification and Description of the Two Case Studies

The two regions chosen as case studies share traits with most MENA nations, including Yemen, Oman, Yemen, Algeria, Morocco, Tunisia, Libya, Algeria, Mauritania, Sudan, Lebanon, Jordan, Iraq, Syria, Kuwait, Bahrain, Saudi Arabia, and Qatar. These nations all face water shortages. The Minia governorate serves as an example of the black soil of older lands in the first study area, whereas the Nubariya region is another example of recently farmed lands with sandy soil. The two research regions are compared in Table 1 and Figure 1. The two areas have similar limited supply resources and elevated seasonal water needs, particularly during the summer.

**Table 1.** Comparison between the two study areas.

| Study Area | Minia | Nubariya |
|---|---|---|
| Location | Central Egypt, Northern part of Upper Egypt, is on the Nile valley's western bank. The province administratively is divided into Nine cities: El-Edwa, Maghagha, Beni-Mazar, Matai, Samalut, El-Minia city, Abo–Qarqas, Mallawi, and Der Mawas, from North to South, respectively, Figure 1. | Located within three Egyptian governorates; Alexandria (Alex. Gov.), Matrouh (Matrouh Gov.) and Behera (Behera Gov.) Governorates, Northern West of Nile Delta, Figure 1. |
| Land use and soil | Part of the old lands, dark soils | Part of the newly cultivated lands, sandy soil |
| Astrological location | Latitudes 27°35′32.39″–28°45′44.04″ N and Longitudes 30°30′51.90″–31°10′0.31″ E | Latitudes 30°28′53.55″–31°5′18.72″ N and Longitudes 28°59′1.22″–30°28′53.55″ E |
| Boundaries | From the North: Bani Souef Governorate, from South: Assuit and Al-Wadi Al-Gadid Governorates, Minia Eastern desert then Eastern desert from the East and Minia Western desert then Al-Giza Governorate from the West | Located between two main roads: Cairo-Alexandria desert road and Wadi Al-Natron- Al-Almeen road (Figure 1). |
| Extension | Extends on Nile River axis from Km 612 to Km 760 North [28], Assiut Barrage controls Nile flow to the study area which are irrigated from Al-Ibrahemia, Bahr Youssef canals and their branches. | It extends along the Mediterranean Sea shore. The study area is almost flat with a very low relief and gently slopes to the Northern East. The altitude varies between 24 and 32 m above mean sea level [29]. |
| Water Resources | Al-Ibrahemia canal upstream of Assiut Barrages on Nile River which carries water to Dairoot center in the North from which Bahr Youssef branches flows to feed the area from the West, Figure 1. | Al-Rayah Al-Nassery, Al-Rayah Al-Behery and Al-Mahmodia canals are the main surface water resources in Alexandria (Alex. Gov.) and Behera (Behera Gov.) Governorates from the Nile River which feed the study area from Nubariya canal through two main canals: Al-Nasr and Mariot canals, Figure 1. |

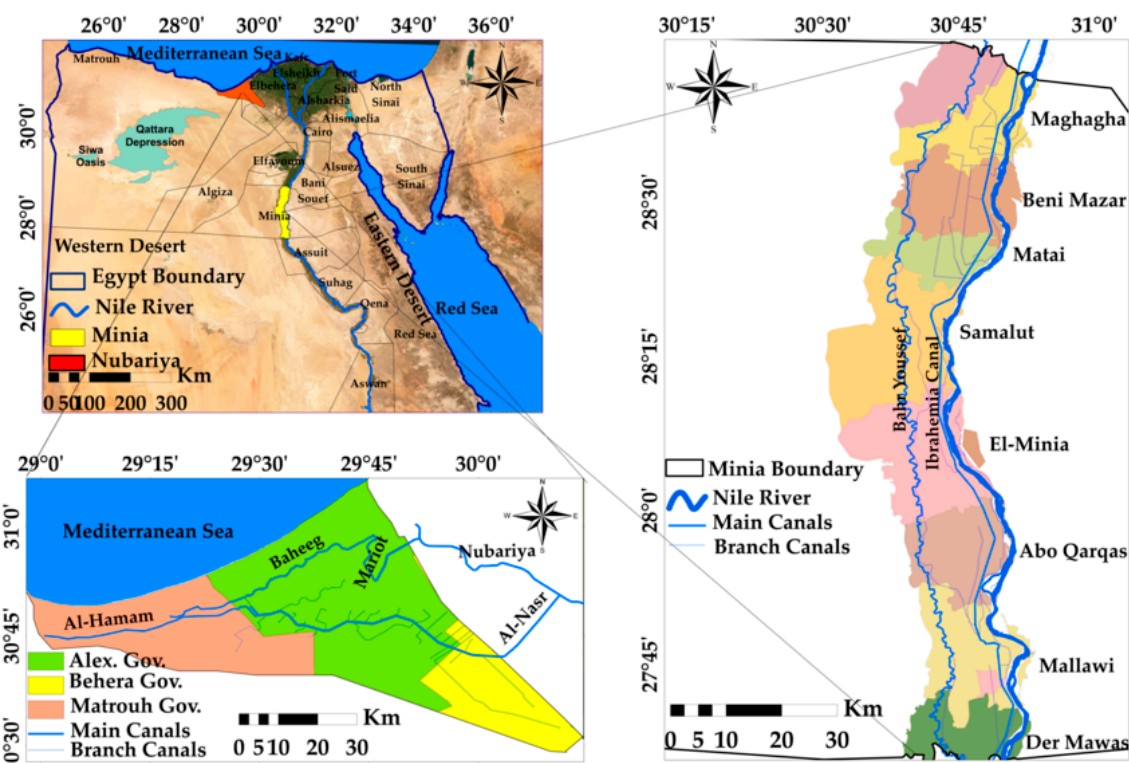

**Figure 1.** Location map of Minia and Nubariya regions.

## 3. Materials and Methods

The Water Evaluation and Planning System (WEAP) (https://www.weap21.org/ (accessed on 24 April 2023)) was used for water resources modeling in Minia Governorate, central Egypt, on the western bank of the Nile Valley, and the Nubariya region, West Nile Delta, Egypt. The concept can be applied to similar urban water supply system patterns that might be anticipated in several cities across the nation, as well as on a worldwide scale. The Minia and Nubariya regions are struggling with dwindling hydrological supplies and rising seasonal water demands, particularly in the summer. Water resources must be properly quantified to create management strategies that are effective in addressing the issue of water scarcity. We attempt to quantify the geographically and temporally defined hydrological balance of the case studies geomorphology to evaluate the sustainability for actual water use.

WEAP models were set up, local main and branch canals were located, and surface flow calibration and validation were all part of the technique. The study's use of surface flows and spring flows involved both measurement-based and estimation-based data, both of which were sourced from the Ministry of Water Resources and Irrigation (MWRI) (https://www.mwri.gov.eg/ (accessed on 27 May 2023). Then, WEAP procedure cases were constructed, parameters had been calculated and examined in accordance with their bounds, compared to determine their effects, and lastly, we evaluated the best case that was the most like the observed flows with accepted parameter values. The model's validations were then examined by contrasting the results from the models' output at known streamflow gauge stations in the two study areas with those obtained from the streamflow simulations. Figure 2 provides a description of the flowchart of used methodology.

The natural system components (rivers) and technical system components (reservoirs, diversions, canals, and cities) are schematized using a network of interconnected model elements with unprojected systems. Model elements are classified into two categories; demand nodes, user-defined supply options, demand priorities, and environmental requirements for each node are what drive the water management concept.

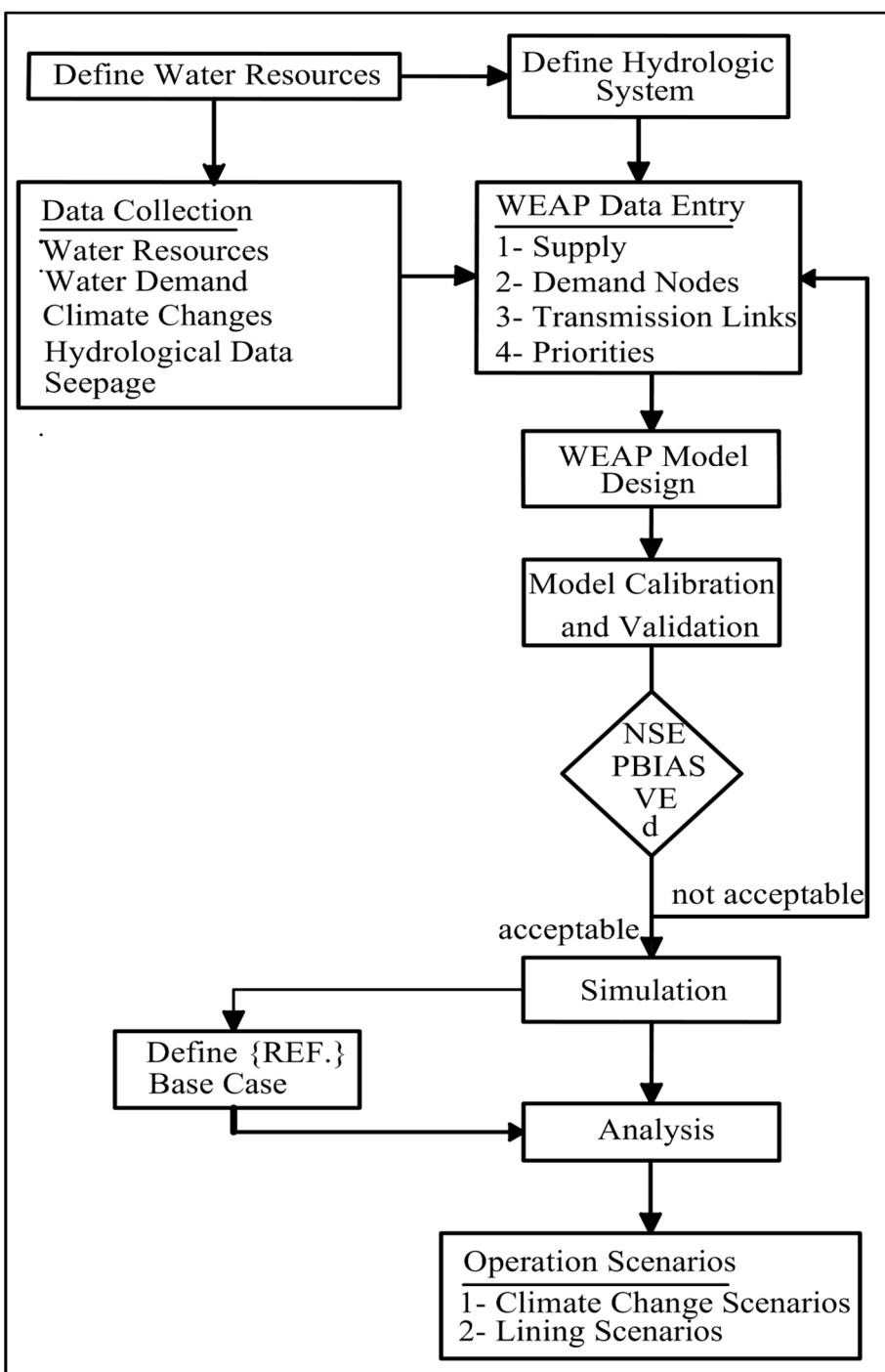

**Figure 2.** Flowchart of the used methodology.

The database used was obtained by drawing the canal streams from Google Earth Pro and Landsat images [30], besides the directories of various sectors, including the MWRI, Ministry of Agriculture (https://moa.gov.eg/en (accessed on 20 April 2023), and Central Agency for Public Mobilization and Statistics (CAPMAS) (https://www.capmas.gov.eg/HomePage.aspx (accessed on 25 February 2023). These data are gathered from surface water and groundwater in the study areas, rains, and reuse mixing stations from agriculture drains.

### 3.1. Data Collection

3.1.1. Water Resources and Demand in Minia

Al-Ibrahemia and Bahr Youssef canals (refer to Figure 1) provide the area with water from Assiut Barrages on the Nile River. Due to the availability of information, 2020 was chosen as the base case. The model was fed by the Al-Ibrahemia Canal discharge value delivered to the study area while considering the minimum flow requirements that must be passed to the Beni-Souef region in the study area's north (Table 2). Groundwater reservoirs are a crucial tool for improving water-use efficiency. Since it was challenging to obtain the groundwater values for each month in the study area, we reduced the total discharge delivered to the study area by the groundwater's total value for the year 2020 as provided by the CAPMAS. The study area also has five primary mixing pump stations for agricultural drainage (Table 3). Table 4 illustrates the monthly values for reusing water from drains through the five mixing stations.

**Table 2.** Ibrahemia canal water discharge (m³/s) during 2020.

| Month | Jan. | Feb. | Mar. | Apr. | May. | Jun. | Jul. | Aug. | Sep. | Oct. | Nov. | Dec. |
|---|---|---|---|---|---|---|---|---|---|---|---|---|
| Minia Region | 117.1 | 247.8 | 234.3 | 285.3 | 358.3 | 380.9 | 399.6 | 391.3 | 322.1 | 316.8 | 236.8 | 191.6 |
| Min. flow to Beni Souef | 23.8 | 38.1 | 43.5 | 48.6 | 60.0 | 68.3 | 71.2 | 71.4 | 69.2 | 66.5 | 41.9 | 32.5 |

**Table 3.** Mixing stations in Minia study area.

| Mixing St. | Source | Destination | No. of Units | Unit Discharge (m³/s) | Data Entry |
|---|---|---|---|---|---|
| Alashmonen | Kabkab drain | Alashmonen canal | 1 | 0.5 | Alashmonen St. |
| Bani Aly | Bani mazar drain | Deromenbal canal | 2 | 1.00–0.5 | Serry br. St. |
| Almessaied | Sakoula | Saba canal | 2 | 0.5 | |
| Belhassa | Magror albahr alyousfy | Belhassa canal | 2 | 1.00–0.5 | Bahr Youssef Stations |
| Abo hashema | Der-alsanforia | Abo hashema canal | 2 | 0.5 | |

**Table 4.** Quantity of water reuse (m³/s) in Minia study area in 2020.

| Month | Jan. | Feb. | Mar. | Apr. | May. | Jun. | Jul. | Aug. | Sep. | Oct. | Nov. | Dec. |
|---|---|---|---|---|---|---|---|---|---|---|---|---|
| Alashmonen | 0.067 | 0.100 | 0.133 | 0.133 | 0.133 | 0.133 | 0.133 | 0.133 | 0.133 | 0.133 | 0.133 | 0.133 |
| Bani Aly | 0.100 | 0.100 | 0.100 | 0.100 | 0.627 | 0.407 | 0.188 | 0.204 | 0.177 | 0.150 | 0.125 | 0.100 |
| Almessaied | 0.108 | 0.108 | 0.108 | 0.108 | 0.328 | 0.214 | 0.100 | 0.174 | 0.158 | 0.142 | 0.125 | 0.108 |
| Belhassa | 0.100 | 0.100 | 0.100 | 0.100 | 0.463 | 0.319 | 0.175 | 0.228 | 0.185 | 0.142 | 0.121 | 0.100 |
| Abo hashema | 0.125 | 0.125 | 0.125 | 0.125 | 0.376 | 0.255 | 0.133 | 0.197 | 0.178 | 0.158 | 0.142 | 0.125 |
| Sum. of reuse | 0.50 | 0.533 | 0.567 | 0.567 | 1.927 | 1.328 | 0.729 | 0.935 | 0.830 | 0.725 | 0.646 | 0.567 |

Rain provides a secondary supply of water in the study area, but its scarcity and variability make it unreliable. The yearly rainfall in the area is expected to be 4 million m³ based on MWRI data. Due to the lack of this quantity and its irregularity, it was subtracted from the total water resource discharge in the area when data were entered into the model during months with reported rainfall events (December to March).

Agriculture is the primary water-consuming sector in the study area, accounting for more than 90% of the total available water resources. The farmed area is around 831,729 feddans and is irrigated by the Nile River and its branches (Ibrahemia and the Bahr Youssef canal). The area served in the model is 792,213 Feddans irrigated from the Ibrahemia and Bahr Youssef canals, as we did not consider the regions irrigated from the Nile in all computations in the study area. Domestic and industrial applications are considered for the second sector, utilizing water through 27 drinking stations (Posters), 7 from Ibrahemia consume about 10,250,368 m³/day, and 20 from Bahr Youssef consume about 37,154,372 m³/day.

3.1.2. Water Resources and Demand in Nubariya

The two main canals that supply the area from the Nile River in the west Nile Delta are the Mariot and Al-Nasr canals, which the Nubariya canal feeds through lifting station

No. 3 because its bed rises uphill from Nubariya canal [29]. We chose 2020 as the study base case based on the data availability for the study area; refer to Figure 1.

Groundwater is an essential resource for reusing irrigation water that has leaked into the soil and hydrogeological layers to boost water efficiency. In the old lands, groundwater is the second most important source of irrigation after Nile water, and it is the primary source in the new lands and reclaimed desert. Groundwater is also a source of safe domestic water. However, in the Alexandria Governorate, rainwater is the second most important source of irrigation after the Nile, as there is no irrigation based on groundwater. The annual average rainfall in the Alexandria Governorate is around 0.12 billion $m^3$/year. It was challenging to obtain monthly groundwater and rainfall values in the study area. As a result, we collected their total values from CAPMAS Information for the year 2020 and deducted them from the total discharge given to the study region during the months when rainfall events were observed (December to March). In addition to one mixing pump station from the Al-Nasr-5 drain in the study area that is lifting the drain's water to the Fara 8 aimen al-Banger branch canal through two pumps with a unit discharge of 1.0 $m^3$/s. The monthly values of water reuse discharge from drains through the mixing station (Mixing St. 18) are listed in (Table 5).

**Table 5.** Al-Nasr and Mariot Canals Water discharge ($m^3$/s) and reuse during 2020.

| Month | Jan. | Feb. | Mar. | Apr. | May. | Jun. | Jul. | Aug. | Sep. | Oct. | Nov. | Dec. |
|---|---|---|---|---|---|---|---|---|---|---|---|---|
| Al-Nasr Canal | 41.67 | 56.05 | 52.99 | 64.56 | 74.32 | 75.81 | 101.62 | 75.41 | 75.55 | 71.72 | 57.52 | 45.78 |
| Mariot Canal | 7.59 | 11.06 | 9.76 | 11.22 | 12.20 | 12.22 | 12.08 | 12.04 | 11.09 | 12.20 | 8.65 | 9.38 |
| Mixing St.-18 | 0.067 | 0.139 | 0.267 | 0.333 | 0.400 | 0.400 | 0.400 | 0.400 | 0.333 | 0.267 | 0.192 | 0.117 |

Agriculture is the third economic sector in the Alexandria Governorate, behind industry and tourism, but it is the dominant economic sector in the Behera Governorate, employing approximately 43% of the Behera workforce. The Nile River and its branches irrigate 522,712 Feddans, the Mariot Canal irrigates 17,100 Feddans, and the Al–Nasr Canal irrigates 505,612 Feddans. In the Nubariya study area, the population is an important sector that consumes water through five drinking water purification stations: three from the Al–Nasr canal consume approximately 61,000 $m^3$/day, one from the Al–Hamam canal consumes approximately 334,000 $m^3$/day, and one from the Mariot canal consumes approximately 640,000 $m^3$/day.

*3.2. Set Up of WEAP Model to the Study Areas*

Using ArcMap vol. 10.3, relevant datasets for the study areas were collected and organized in a geodatabase. GIS tools were used to process geospatial data. Hydrogeology, land use, water supply, agriculture demand, domestic and industrial consumption, mixing stations from drains, and management practices are among the organized data. WEAP was utilized to schematize the river network, canals, groundwater aquifers, and streams using GIS maps as a reference background. Water resources were classified into three categories: (1) dynamic bodies (such as surface water); (2) static bodies (such as groundwater and reservoir water); and (3) demand bodies (such as cities and agriculture). Following the identification of these forms, the allocation of water resources for the year 2020 was illustrated.

Minia schematization led to 29 demand nodes, 3 drainage reuse nodes, and 4 poster sites for domestic use, as shown in Figure 3. Nodes for demand sites were added in each municipality dealing with livestock, domestic, industrial, and tourism water use.

To accommodate water exchanges between the environment and the water sectors, all nodes were linked with 39 transmission links and return elements. The rainfall modeling approach was based on MWRI data, which was subtracted from the overall water supply to the study area.

On the other hand, Nubariya schematization resulted in 25 demand nodes, 1 reuse node, and 3 poster sites for domestic use (Figure 4). In each municipality, demand sites

for livestock, household, industrial, and tourism water usage were added. To account for water transfers between the environment and the water sectors or across the various water sectors, all nodes were linked with 26 transmission and return elements. The rainfall volume in Nubariya was calculated using MWRI data and deducted from the research area's total water supply.

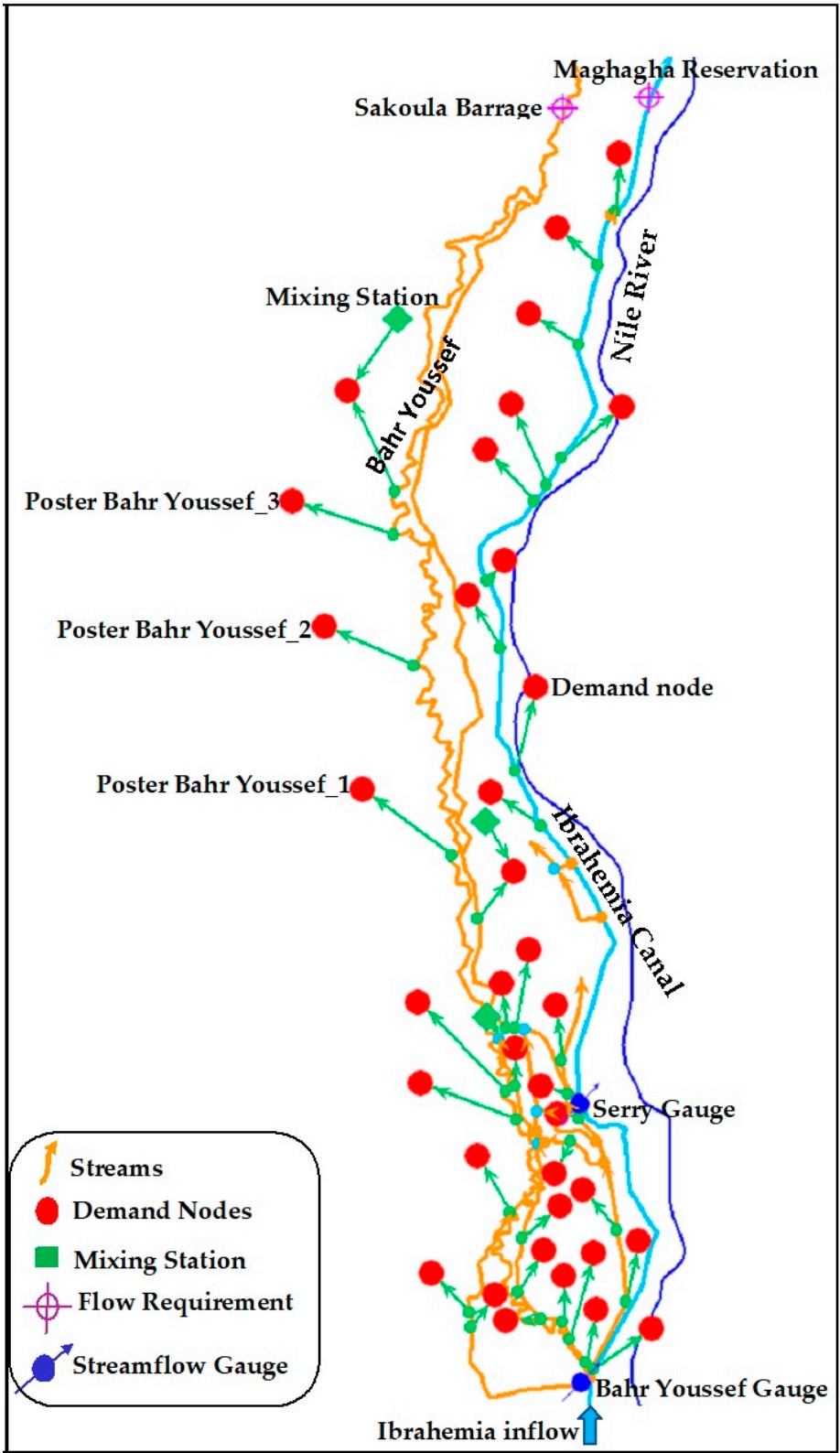

**Figure 3.** Schematic diagram showing flow network and demand site locations for Minia.

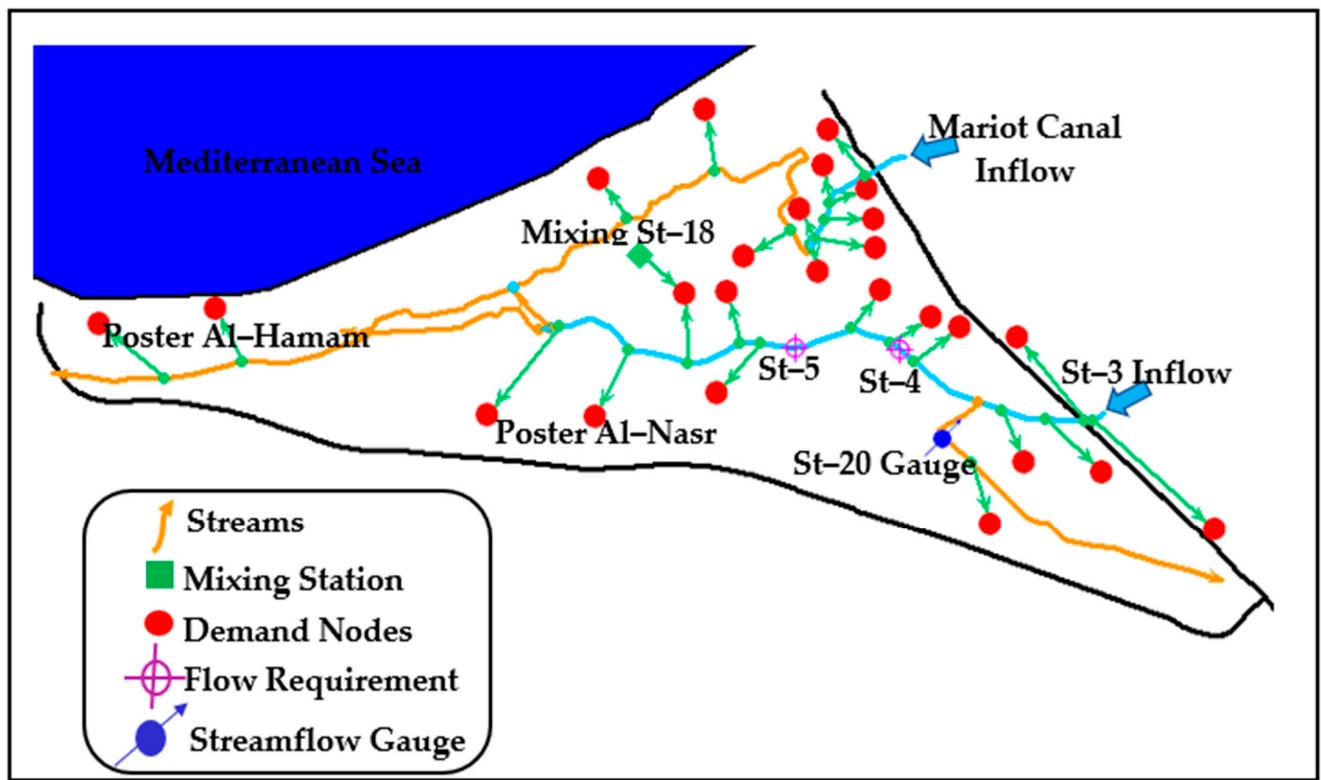

**Figure 4.** Schematic diagram showing flow network and demand site locations for Nubariya.

WEAP achieves the principle of water balance for every link and node in the flow network according to demand priorities, supply preferences, mass balance, and other constraints. Mass balance equations are the WEAP foundation, where monthly water accounting: total inflows minus total outflows equals the net change in storage, if any.

Annual Demand in WEAP: The monthly demand represents the needed amount of water for the monthly site's demand. A demand site (DS) for water is calculated as the sum of the demands for all demand sites through the smallest level branches [25].

$$\text{Annual Demand} = \text{Total Activity} * \text{Water Use Rate} \qquad (1)$$

The annual demand is calculated according to crop patterns in the study areas obtained from CAPMAS, and the consumptive use for each crop depends on the Penman equation, where the seasonal or monthly consumptive water use (*Et*) of a crop, in inches is [31]:

$$Et = K_c \times \sum F \qquad (2)$$

$$F = \frac{(T \times \rho)}{100} \qquad (3)$$

where (*F*) is the monthly water consumption factor, (*T*) is the mean temperature through the month in Fahrenheit, and (*ρ*) is the monthly percentage of daytime. The crop coefficient $K_c$ is an empirical seasonal factor relating the seasonal plant water usage for a specific crop to the total seasonal consumptive use of water generated. To calculate $K_c$, the actual crop requirement of irrigation water must be measured during the growing season [32–34], $K_c$ can be calculated from measured *F* and *Et*. The share of each crop was calculated according to crop pattern consumptive use, and the cultivated area in the season. Then, monthly variation was calculated for each month in the study areas, as expressed in Equations (4) and (5).

$$MSh_i = Cu\% \times CA \qquad (4)$$

$$M \cdot Var = \frac{\sum_{i=1}^{i=12} MSh_i}{\sum area} \tag{5}$$

where $MSh_i$ is the share of consumptive use for each month during the year 2020, $Cu\%$ is the consumptive use percent for each crop, $CA$ is the crop cultivated area (Feddans), $M \cdot Var$ is the monthly variation percent, and $\sum area$ is the summation of total cultivated areas in the region during 2020 (in Feddans).

Monthly requirement from supply node: The supply requirement is the actual water amount that is needed from the supply sources. The supply requirement takes the demand and modifies it to account for internal reuse, demand site management strategies for reducing demand, and internal losses [25,35]. These three adjustment fractions are part of the input data in the model.

$$\text{Monthly Supply Req.} = \frac{\text{Monthly Demand} \times (1 - \text{Reuse Rate}) \times (1 - \text{DSM Savings})}{1 - \text{Loss Rate}} \tag{6}$$

Transmission Links: Transmission links deliver water from supply nodes for final demand satisfaction at demand sites. In addition, they deliver drainage and wastewater outflows from demand sites for reuse [36]. The amount delivered to the demand site equals the amount withdrawn from the source minus any losses. Losses indicate the evaporative loss and a percentage of the flow passing through a transmission link.

$$\text{Trans. Link Outflow} = \text{Trans. Link Inflow} - \text{Trans. Link Loss} \tag{7}$$

Referring to the WEAP model setup (Figures 3 and 4 for the region schematics), flow requirements and demand sites were allocated water according to demand priorities. When there is a shortage of water, demand priorities are beneficial in illustrating the systems of water rights and in ensuring that higher priorities are met as completely as possible before smaller needs are considered [17,37]. In WEAP, the priority index ranges from 1 to 99, with the lower value representing higher priority and vice versa. Meanwhile, when priorities were the same, shortages were equally shared among all the demands.

### 3.3. Calibration Procedures

The complexity of water allocation models and the fact that they are required for human behavior simulation to reflect demand changes besides physical processes means that model calibration and validation are particularly challenging and have historically been disregarded [35,38,39]. Calibration of a model refers to changing the values of the parameters so that the simulated results closely match the observed data [40]. For the two models' calibration, observed stream flow data were obtained from two gauging stations located on the main canals in the study areas, beside the main head flows to the areas. Calibration included changing the model parameters for better base-case simulation. WEAP21 has no routine for automatic calibration, so the changes implemented were tested by comparing the observed and simulated flows at the three observed points.

To calibrate the models, three calibration procedures were used to obtain the best case that satisfied the most acceptable parameters accuracy. The calibrated procedures are categorized by cases 1, 2, and 3.

Referring to Figure 3, the two gauging sites in Minia are Sakoula Barrage at the end of the Bahr Youssef canal, and Maghagha Reservation at the end of Ibrahemia canal.

First, the model was performed in the base situation (case 1) using only the main head flow coming into the area from the Ibrahemia Canal (Table S1 and Figure S1 in the Supplemental Materials). The model was then run with the observed discharge for the second gauge site from the chosen sites, Maghagha Reservation, which was obtained from MWRI. The results are shown in Table S2 and Figure S2 in the Supplemental Materials as values for the simulated flows (case 2). The model was then run using the observed

discharge for the third gauge site, Sakoula Barrage, and the results were obtained as stream flow values (case 3), as shown in Table 6 and Figure 5.

**Table 6.** Minia case 3 calibrated flows (Mm$^3$/month) obtained from the WEAP model, showing data entry for Ibrahemia, Maghagha and Sakoula barrages inflows.

| Month | | Jan. | Feb. | Mar. | Apr. | May. | Jun. | Jul. | Aug. | Sep. | Oct. | Nov. | Dec. |
|---|---|---|---|---|---|---|---|---|---|---|---|---|---|
| Ibrahemia inflow | Simulated | 307.49 | 626.18 | 615.97 | 739.64 | 941.30 | 965.54 | 1049.49 | 1027.69 | 835.07 | 832.48 | 613.85 | 485.09 |
| | Observed | 303.67 | 642.49 | 607.54 | 739.65 | 928.93 | 987.54 | 1035.85 | 1014.47 | 835.08 | 821.29 | 613.85 | 496.70 |
| Maghagha Reservation | Simulated | 63.98 | 115.04 | 116.66 | 134.07 | 175.43 | 177.25 | 190.90 | 191.34 | 179.52 | 178.14 | 108.80 | 93.96 |
| | Observed | 61.91 | 102.87 | 112.89 | 126.19 | 155.62 | 177.24 | 184.74 | 185.17 | 179.52 | 172.39 | 108.80 | 84.40 |
| Sakoula Barrages | Simulated | 175.21 | 295.11 | 257.16 | 291.81 | 395.98 | 400.81 | 393.80 | 376.69 | 326.96 | 336.01 | 243.20 | 227.02 |
| | Observed | 175.21 | 279.74 | 257.16 | 275.39 | 360.95 | 400.83 | 393.77 | 376.72 | 326.88 | 336.02 | 243.10 | 208.35 |

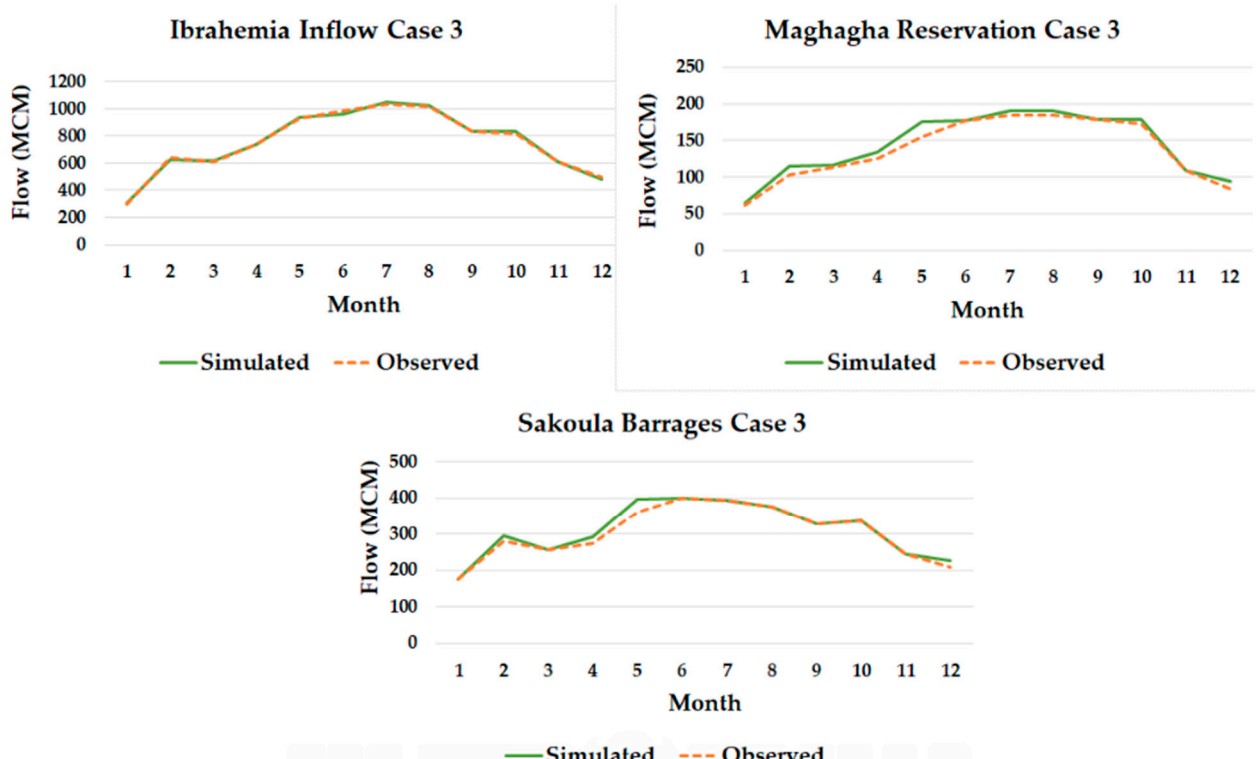

**Figure 5.** Case 3 calibrated flows obtained from the WEAP model in Minia.

In each instance, model capabilities were investigated by comparing the simulated flow results with the actual flows in the area.

On the other hand, the two gauging sites in Nubariya are (1) station number 4 (St–4) and (2) station number 5 (St–5), as show in Figure 4.

First, the model was performed in the base situation (case 1) using only the main head flow coming into the area from station number 3 on Al–Nasr Canal (St–3), as shown in Table S1 and Figure S1 in the Supplementary Materials. The model was then run with the observed discharge for the second gauge site from the chosen sites, station number 4 (St–4), which was obtained from MWRI. The results are shown in Table S4 and Figure S4 in the Supplemental Materials as values for the simulated flows (case 2). The model was then run using the observed discharge for the third gauge site, station number 5 (St– 5), and the results were obtained as stream flow values (case 3), as shown in Table 7 and Figure 6.

**Table 7.** Nubariya case 3 calibrated flows (Mm³/month) obtained from the WEAP model, showing data for St–3, St–4, and St–5 inflows.

| Month | | Jan. | Feb. | Mar. | Apr. | May. | Jun. | Jul. | Aug. | Sep. | Oct. | Nov. | Dec. |
|---|---|---|---|---|---|---|---|---|---|---|---|---|---|
| Al-Nasr Canal Discharge (St.3) | Simulated | 111.61 | 140.44 | 141.93 | 167.34 | 199.06 | 196.50 | 272.18 | 201.98 | 195.82 | 192.09 | 149.09 | 122.62 |
| | Observed | 108.0 | 145.28 | 137.35 | 167.33 | 192.64 | 196.50 | 263.41 | 195.45 | 195.81 | 185.88 | 149.1 | 118.66 |
| Al-Nasr Canal Discharge (St.4) | Simulated | 76.14 | 86.68 | 92.93 | 104.41 | 115.43 | 101.83 | 110.84 | 124.27 | 109.52 | 107.64 | 91.49 | 76.54 |
| | Observed | 72.04 | 84.95 | 87.59 | 100.38 | 109.48 | 98.60 | 107.27 | 120.72 | 103.38 | 101.71 | 83.12 | 71.96 |
| Al-Nasr Canal Discharge (St.5) | Simulated | 65.55 | 84.01 | 81.32 | 88.76 | 120.20 | 100.21 | 95.10 | 105.64 | 100.0 | 120.30 | 87.70 | 78.00 |
| | Observed | 68.74 | 85.76 | 78.69 | 87.67 | 97.39 | 100.19 | 100.81 | 102.21 | 99.99 | 99.22 | 86.70 | 64.89 |

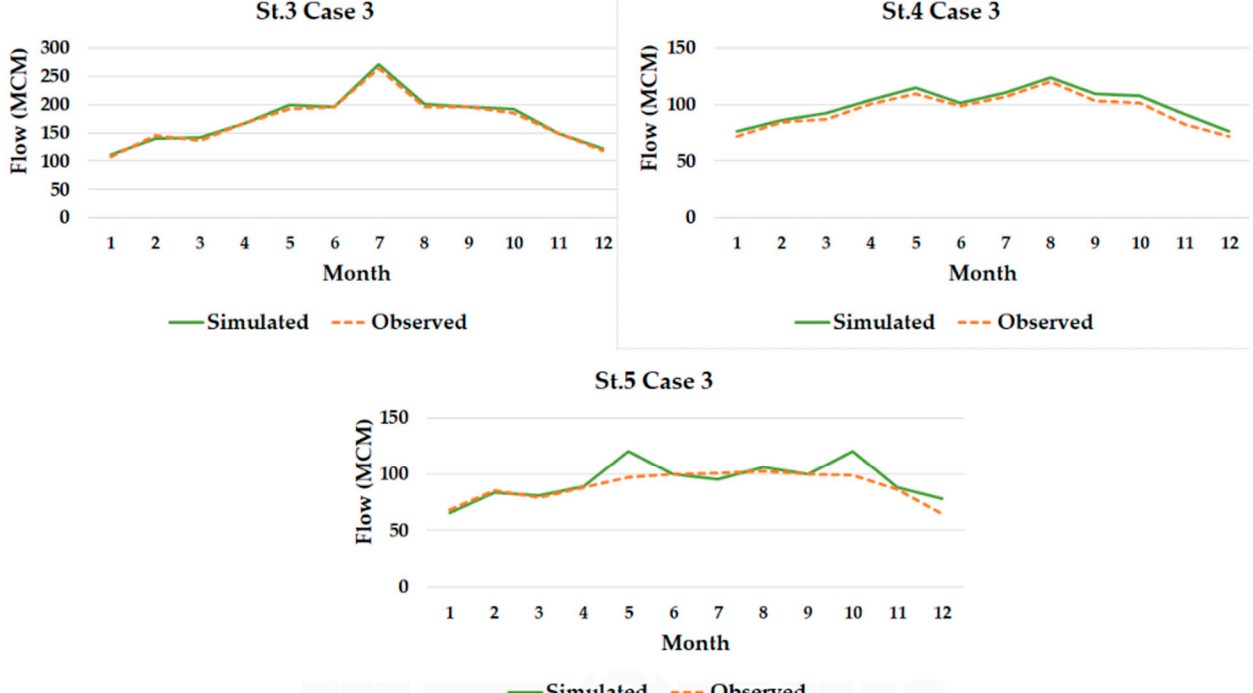

**Figure 6.** Case 3 calibrated flows obtained from the WEAP model in Nubariya.

In each instance, model capabilities were investigated by comparing the simulated flow results with the actual flows in the area.

Two types of studies were performed for the model's calibration: a comparison of the monthly and annual flows and a comparison of the flow frequencies. Physical observation serves as the basis for calibration [41]. The comparison can be made for downstream stations where canal flows are calculated by simulation, with head flows directly entering the WEAP system. The method consists of a comparison (monthly or year-by-year) between the simulated and observed flows.

To evaluate the performance of the calibration results, the statistical parameters considered for this study were percent Bias (*PBIAS*), Nash–Sutcliffe efficiency (*NSE*), agreement index (*d*), and volumetric efficiency (*VE*), as expressed below [26,42,43].

### 3.3.1. Percent Bias (PBIAS)

The average simulated data's tendency to be greater or smaller than the corresponding real items is measured by percent bias (*PBIAS*) [26,42–44]. Low *PBIAS* levels signify accurate model simulation, and the ideal value is zero. Positive values indicate an underes-

timating bias in the model, whereas negative values indicate an overestimation bias in the model [40,43,44]. PBIAS is calculated with Equation (8)

$$PBIAS = 100 \times \frac{\sum_{i=1}^{n}(O_i - S_i)}{\sum_{i=1}^{n} O_i} \tag{8}$$

where $O_i$ is the $i$th value of the observed data, $S_i$ is the $i$th simulated value, and $n$ is the total observations number.

### 3.3.2. Nash–Sutcliffe Efficiency (*NSE*)

The two main justifications for the suggestion of *NSE* are that (1) ASCE, 1993 [45,46] recommends its use and (2) it provides a variety of information and frequently applies to described values. It should also be noticed that *NSE* is the optimal objective function, reflecting the overall acceptance of a hydrograph [47].

Nash–Sutcliffe efficiency (*NSE*) is an indicator that is normalized to determine the relative magnitude of the residual variance compared to the observed data variance [26,43,48,49]. *NSE* reflects how well the plot of observed data compared to simulated data fits the 1:1 line. *NSE* is computed as follows in Equation (9)

$$NSE = 1 - \frac{\sum_{i=1}^{n}(O_i - S_i)^2}{\sum_{i=1}^{n}(O_i - \overline{O})^2} \tag{9}$$

where $O$ is the mean of the observed data. *NSE* ranges between $-\infty$ and 1.0, with *NSE* = 1 being the ideal value. Values between 0.0 and 1.0 are generally viewed as acceptable performance levels, while values <0.0 indicate that the prediction of the mean observed value is better than the simulated value, which indicates an unacceptable level of performance [26,42,49].

If *NSE* has greater values, the better the model reproduces observations, with 1 being the ideal situation [49,50].

### 3.3.3. Agreement Index (*d*)

The degree of error for model prediction is uniformly measured by the index of agreement (*d*) [51] as a uniform measure of the degree of error for model prediction and varies between 0.0 and 1.0. Between the measured and simulated values, a computed value of 1 shows perfect agreement, while a calculated value of 0 indicates no agreement at all [51,52]. The agreement index signifies the potential error ratio and the mean square error [53,54]. The summation of the squared absolute values of the distances from the mean of measured observed values to the simulated values and the distances from observed values to the mean of observed values is used to quantify the potential error. The *d* can detect both relative and additive differences in the observed and simulated variances and means; however, *d* is too sensitive to extreme values due to the squared differences [41,46]. *d* is computed as shown in Equation (10) as follows:

$$d = 1 - \frac{\sum_{i=1}^{n}(O_i - S_i)^2}{\sum_{i=1}^{n}\left(\left|S_i - \overline{O}\right| + \left|O_i - \overline{O}\right|\right)^2} \tag{10}$$

### 3.3.4. Volumetric Efficiency (*VE*):

According to [26,42,49], generally, if *VE* is $0 - 1$, the model performance is satisfactory, with 1 being perfect agreement between observed and simulated values. *VE* is calculated as shown in Equation (11):

$$VE = 1 - \frac{\sum_{i=1}^{n}|S_i - O_i|}{\sum_{i=1}^{n}(O_i)} \tag{11}$$

*3.4. Model Validation*

Validation is the process by which we run the model with an independent set of data and compare the simulated results with the observed data. If the simulated results closely match the observed data, then the model is validated. In Minia, to validate the model, observed stream flow data obtained from two streamflow gauging stations located on the main canals in the study area, the flows of Bahr Youssef and Serry gauges were compared to their model results. Consequently, in Nubariya, to validate the model, observed stream flow data obtained from station (St–20) which is located on Fara–20 canal in the study area, was compared to its modeled results (refer to Figures 3 and 4).

*3.5. Operation Scenarios*

Based on the water resources and the metro-hydrological conditions in the two study areas, we tested the following scenarios built on the REF base case we obtained from the validation step:

- Two main scenarios (SCI and SCII) lead to five sub scenarios being examined for estimating the water deficit amount and improving water use efficiency in the two study areas (Table 8). The two scenarios were discussed according to historical climatic data and the Egyptian water strategy, and Egypt's National Water Resources Plan (NWRP, 2050).

**Table 8.** Operation scenarios using WEAP in Minia and Nubariya.

| Operation Scenarios | | | |
|---|---|---|---|
| CL. Change (I) | I–1: Past (1960–1990) | I–2: Current (1991–2020) | I–3: Predictive (2021–2050) |
| Lining (II) | II–1: Lining of 10% canals length | II–2: Lining 25% of canals length | |

3.5.1. Climate Change Scenarios Justification "Cl. Change (I)"

Today, climate change is the world's biggest environmental challenge. Developing nations must adopt solutions to many climate change challenges [55]. With a changing climate, the hydrological non-stationarity challenge is new [56].

Climate change is predicted to greatly impact many aspects of human society, from water supply to agriculture and energy production [10,57–59]. Climate change has been identified as one of the world's most important concerns. Global temperatures have risen by around 1 °C from preindustrial times as a result of greenhouse gas emissions induced by human activities, such as methane and carbon dioxide ($CO_2$) [60,61].

Because of the large and growing Nile River population, climate change's impact on Egypt's water resources can be considered one of the most pressing challenges. Egypt has already reached the poverty limit for water [61]. Egypt is very vulnerable to the effects of climate change, according to a 2002 assessment by the Egyptian Environmental Affairs Agency (EEAA), which stated that "due to the dense population, if climate change makes the climate in Egypt drier or warmer, pressures would become intensive on agriculture." [62]. Changes in the Nile's flow, which provides irrigation water for agriculture, would undoubtedly have an impact on Egypt's economy. The construction of additional dams on the Nile River is one way that climatic and regional change could reduce the flow of the Nile [2]. As a result, our study has concentrated on crop irrigation water because agriculture is Egypt's major use of water.

In any water body or natural area, evapotranspiration (*ET*) is one of the significant outcomes in the water balance equation. The Thornthwaite [63] method is widely used for estimation of potential (*ET*) [64]. As a significant method based on average monthly temperature, potential (*ET*) estimation was adopted in the two study areas depending on the Thornthwaite method. Egypt has 31 stations with climatic daily temperature measurements [65]. While each study area has one meteorological station from the 31 stations in Egypt, including Minia Climate St. and Alex. Climate St., as shown in Figure 7.

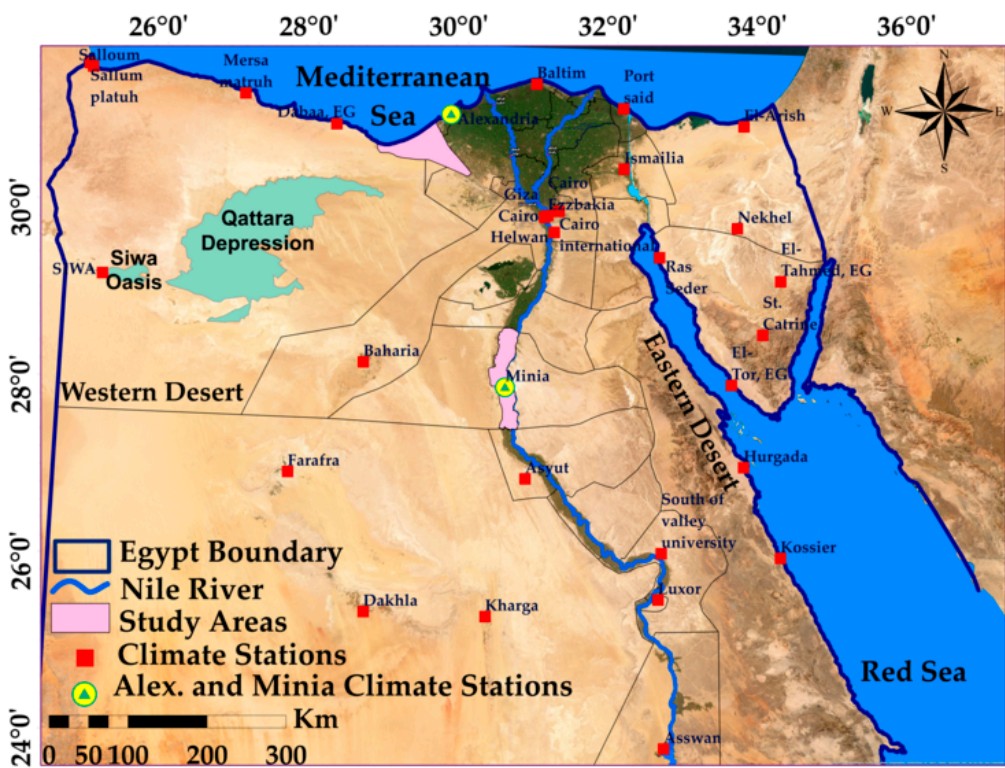

**Figure 7.** Climate stations in Egypt and in the study areas.

Meanwhile, if temperatures increase, it will cause an increase in (*ET*) causing an increase in water demand [61,66]. Temperature data were updated so that continuous time series were obtained. Figure 8 depicts the average temperatures at the two climate stations in 2020 (https://www.ncdc.noaa.gov (accessed on 24 March 2023)).

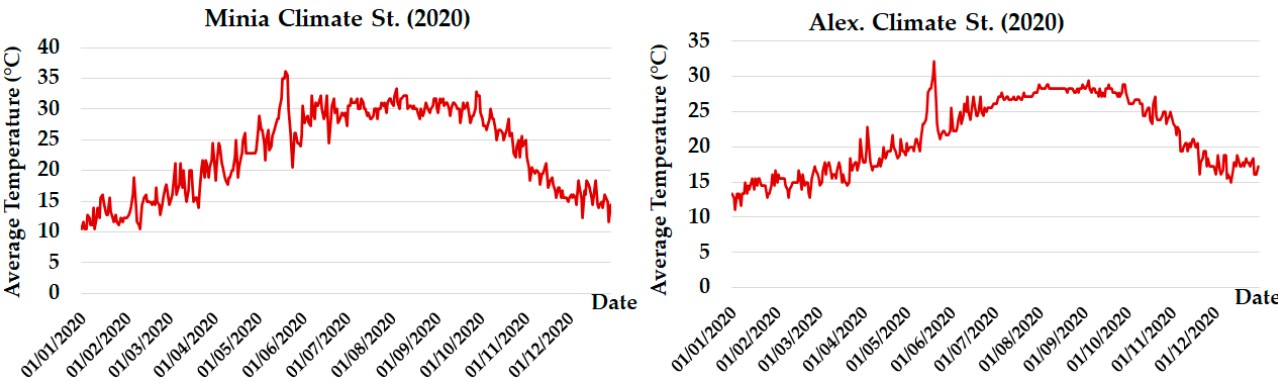

**Figure 8.** Average daily temperature in the study areas in 2020.

Three climatic scenarios were projected to represent the past, current, and future (predicted) conditions. Scenario I–1 represents the past conditions throughout the period (1960–1990). Scenario I–2 represents the current situation through the period (1991–2020) of climatic conditions. Based on the obtained regression lines pathways, scenarios I–3 were predicted (2021–2050). As water requirements for crops are the major component of the water balance in Egypt. Thus, updating the crop requirements is essential and important when considering all climatic-dependent (*ET*), agroclimatic regions, and crop

coefficients [67]. Based on the monthly meteorological data, we estimated potential (*ET*) using Thornthwaite's equation using the following formulae [64]

$$PE = 16 \left[ \frac{10t_n}{J} \right]^a \tag{12}$$

$$J = \sum_{1}^{12} j \tag{13}$$

$$j = \left[ \frac{t_n}{5} \right]^{1.514} \tag{14}$$

$$a = 0.016J + 0.5 \tag{15}$$

$$ET = k \times PE \tag{16}$$

where *k*: is a constant to correct *PE* for latitude other than $0^0$. *PE*: potential evapotranspiration, *J*: Heat Index, *j*: Coefficient monthly temperature (°C), *a*: Constant, and $t_n$: Average monthly temperature (°C).

### 3.5.2. Canals Lining Scenarios Justification (II)

Rehabilitation is a stream management strategy that has been used all throughout the world, including Egypt [68]. The process of transferring and distributing water across open canal networks is the main cause of irrigation water loss. Due to Egypt's restricted river system, there is still not enough water available to meet demand [69]. Egypt is currently dealing with a major difficulty in filling the enormous gap between the required and available irrigation water. Additionally, this is the justification for the quick national project implementation for irrigation canal lining and restoration across the Egyptian countryside to meet the rising demand for food and agricultural productivity, as well as deal with the issues associated with rapid population expansion, as well as to preserve the large amount of wasted seepage water from the canals cross-section to the permeable soil in which they were excavated [70–74]. Without governorate borders, the total length of irrigation canals in Egypt exceeds 33,500 km within 20 governorates [75]. It is expected to gain and save a yearly amount of water of about 5 (BCM), which is a very large quantity for such a great national project. Other valuable gains are also expected in the environment and social sectors, including health.

Rehabilitation is the process by which a waterway's structural integrity can be restored to a good working or previously degraded condition, as shown in Figure 9 [68,76,77]. According to [68,70,78,79], we can gain a reduction in seepage by about 60–80% by using concrete linings, but possible cracks or ill-constructed joints can cause seepage losses. In the current study areas, concrete lining has already been used. There are many equations used to estimate the amount of seepage water depending on soil type, length, wetted perimeter, water depth, width, hydraulic radius, slope, velocity, and discharge.

Furthermore, due to its high cost [70], the canal's concrete rehabilitation badly affects groundwater quality [80]. One of its negative effects is the decreasing groundwater level [81], as well as some factors that affect the use of concrete, including external load, temperature, and humidity [72,82]. After rehabilitation, these factors may cause surface cracks; in sequence, they cause no differences between seepage losses from rehabilitated and unrehabilitated canals with the expansion of cracks on the canal surface [70,82].

To study seepage water losses and the saved water values from lining we selected two different canals in the two study areas, Oshroba canal was studied in Minia, where *b* (*bed width*) = 2.00 m; *S* Seepage losses (m³/s); *C* (Constant depends on the soil properties); Canal length, *L* = 9.260 Km; wetted perimeter, *P* = 4.828 m; and Hydraulic radius,

$R = 0.613$ m. Additionally, the Abo-Massood Al-Yomna canal was studied in the Nubariya Region, where its length was $L = 9.00$ Km and its bed width was $b = 2.00$ m.

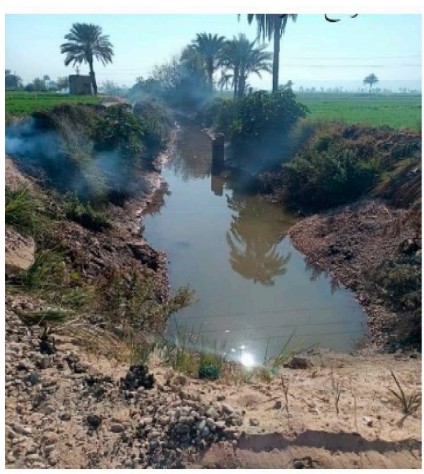 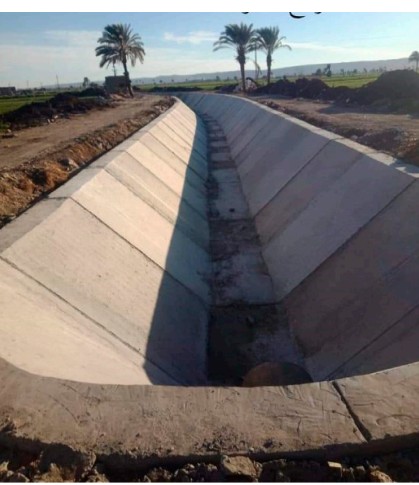

**Figure 9.** Difference between the existing distorted and the designed sections [79].

## 4. Results and Discussions

### 4.1. Models Calibration and Validation

To evaluate the calibration performance results, the statistical parameters: agreement index (*d*), Nash–Sutcliffe Efficiency (*NSE*), Percent BIAS (*PBIAS*), and Volumetric Efficiency (*VE*) were computed for the procedures of all cases. The observed and WEAP-simulated flow hydrographs were compared, and their fitness was physically seen.

WEAP models for the Minia and Nubariya regions were calibrated for the year 2020. The PBIAS calibration results varied between −4.44 and −0.14 in Minia and varied between −9.07 and −1.71 in Nubariya, while NSE was 0.99 in whole Minia gauges and varied between 0.95 and 0.99 in Nubariya. On the other hand, d varied from 0.98 to 0.99 in the two case studies. Calibration gauges and *VE* varied between 0.96 and 0.99 in Minia and between 0.91 and 0.98 in Nubariya calibration gauges. The calculated parameters indicated that there is a very good agreement between the observed and simulated hydrographs in case 3 of the gauging sites in the two regions, Minia and Nubariya.

By comparing the results of the different cases (1, 2, and 3) it is obvious that all statistical parameters are found to be "very good" in general, with the general performance rate in case 3 used for calibration in the two case studies (Table 9). The calculated values were (*PBIAS* is ≤10%, 0.75 < *NSE* < 1.0, the values *d*, and *VE* range between 0 and 1) for calibrated sites in the study areas as indicated in (Table 9) for Minia and Nubariya. This indicates that the models' simulations reasonably agree with the observed data and proves the model suitability for hydrological simulation in the selected case studies.

Validation results for the gauge stations in the two case studies showed acceptable values for PBIAS in the selected validation gauges in Minia, −1.56 and −6.52, as well as in Nubariya, −3.45. The *NSE* and *d* validation values were accepted values of 0.98 and 0.99, respectively. VE were 0.93 and 0.98 in Minia and 0.97 in Nubariya validation sites. The results showed that the maximum discharge in Bahr Youssef was approximately 15.5 million cubic meters per day (Figure 10), as mentioned by [68]. In addition, the results for the validation site in Nubariya showed a very good agreement between the simulated and observed results. Generally, the simulated results for the validation sites reasonably well agree with the observed data. Performance statistics of the validation results are shown in (Table S5) in the Supplementary Materials. The statistical parameters are within desirable ranges for all gauges compared with their general performance ratings, as illustrated in Table 10.

**Table 9.** Performance statistics for calibration and validation results in the two case studies.

| Study Area | Gauges | Statistics of Calibration Results | | | |
| --- | --- | --- | --- | --- | --- |
| | | PBIAS | NSE | d | VE |
| Minia | Ibrahemia inflow | −0.14 | 0.99 | 0.99 | 0.99 |
| | Maghagha Reservation | −4.44 | 0.99 | 0.99 | 0.96 |
| | Sakoula Barrage | −2.35 | 0.99 | 0.98 | 0.98 |
| Nubariya | St.3 | −1.71 | 0.99 | 0.99 | 0.98 |
| | St.4 | −7.15 | 0.98 | 0.99 | 0.93 |
| | St.5 | −9.07 | 0.95 | 0.98 | 0.91 |
| Study Area | Gauges | Statistics of Validation Results | | | |
| | | PBIAS | NSE | d | VE |
| Minia | Bahr Youssef gauge station | −6.52 | 0.98 | 0.99 | 0.93 |
| | Serry Canal gauge station | −1.56 | 0.99 | 0.99 | 0.98 |
| Nubariya | St.20-gauge station | −3.45 | 0.99 | 0.99 | 0.97 |

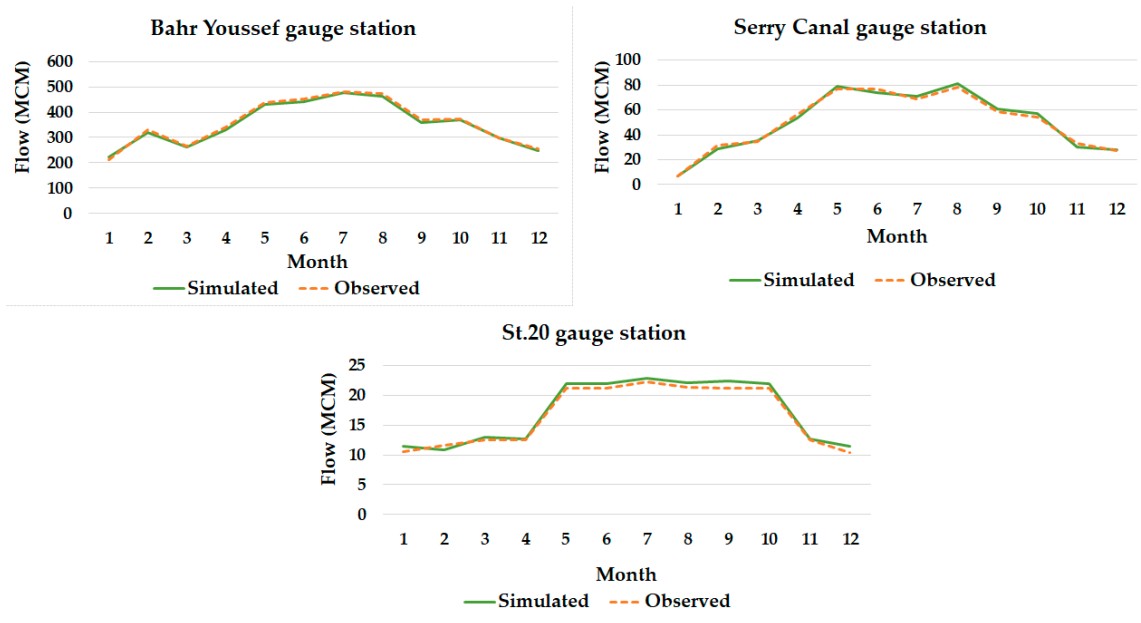

**Figure 10.** Validated flow obtained from WEAP model for Minia and Nubariya.

**Table 10.** General performance ratings for recommended statistics for a monthly time step [42].

| Performance Rating | NSE | PBIAS (%) | VE | d |
| --- | --- | --- | --- | --- |
| Very good | 0.75 < NSE < 1.00 | PBIAS < ±10 | 0–1 | 0–1 |
| Good | 0.65 < NSE < 0.75 | ±10 < PBIAS < ±15 | | |
| Satisfactory | 0.50 < NSE < 0.65 | ±15 < PBIAS < ±25 | | |
| Unsatisfactory | NSE < 0.50 | PBIAS > ±25 | | |

From the presented results in Table 9 and their study, we found that the models are accurately capable of predicting water demand in the study areas, with average accuracy up to 95.5% in Minia and up to 97% in Nubariya.

The calibration procedures are summarized as the Reference Base Case (REF) which represents the actual current situation and is modeled, and projected under current operation conditions in the year 2020. The REF Base Case was created and used to incorporate current identifiable trends in development, water-use efficiency, water supply availability, and other aspects. At the data-view platform in WEAP, each demand site's water consump-

tion was calculated on the REF base case by multiplying the overall activity level by the rates of water use, based on the monthly variation of each site's demand.

From the observed water discharge values that have been obtained from MWRI, the simulated water shortage through the year 2020 in the two case studies can be briefed as 474.13 and 227.81 million cubic meters in Minia and Nubariya, (Figure 11), which show logical data based on the personal data we obtained from the director of the General Directorate of Water Distribution, MWRI.

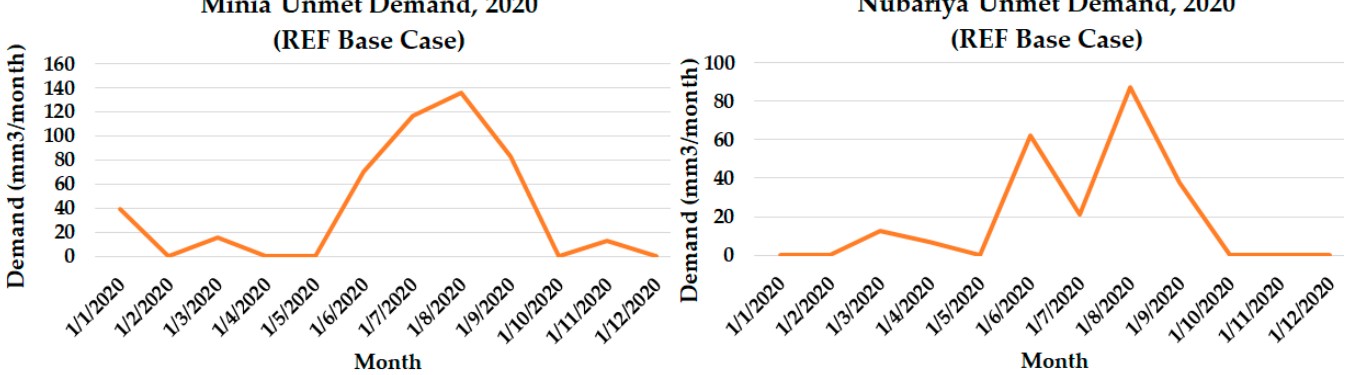

**Figure 11.** REF base case unmet demand in Minia and Nubariya.

The results show that the maximum shortage of water occurs in the summer, from June to September. In Minia, the maximum water consumption was due to maize and sugar cane, which consumed about 59% of the total water consumption, while in Nubariya, the maximum water consumption was due to rice and maize plants, which consumed about 50% of the total water consumption in the year 2020.

Model results in Minia showed water shortages in Bahr Youssef branches, as it is a natural canal passing through four governorates in Egypt: Assuit, Minia, Beni-Souef, and Fayoum. Its regime has a lot of erosion problems, meanders, and sedimentation. Four controlling barrages are located on its waterway and affect its regime. They rule its watercourse and affect its system, leading to erosion and sedimentation. Therefore, it reduced the water's cross-section. Consequently, this decreased carrying capacity for the discharge of the required water demand [68]. On the other hand, model results in Nubariya showed the shortage in the Ayser al–Banger canal, as its area is not served by modern irrigation, as is the case for the entire study area (based on unpublished data obtained from the director of the Al–Nasr General Directorate of Water Resources and Irrigation).

### 4.2. Climate Change Scenarios

The actual *ET* was calculated based on the crop water requirements for the two study areas. According to the application of Thornthwaite's equation [63] in the two scenarios (past and current), in Minia, *ET* increased from 1960 to 1967, decreased in 1968, then increased to 1975. Additionally, it decreased from 1975 to 1980 and returned to increasing from 1980 to 1985, while decreasing until 1987. It decreased from 1987 to 1990, returning to an increase from 1990 to 1995, then decreased in 1996 and 1997, returning to an increase in 2000, and so on. This cycling behavior of the *ET* in the two areas can be observed over the studied years (Figure 12), which is attributed to the variation of climate change and climate variables.

From the analysis of regression lines in the studied zones and according to the application of Thornthwaite's equation [63], statistical values were calculated for temperature and *ET* for past, current, and predictive scenarios as follows (Table 11):

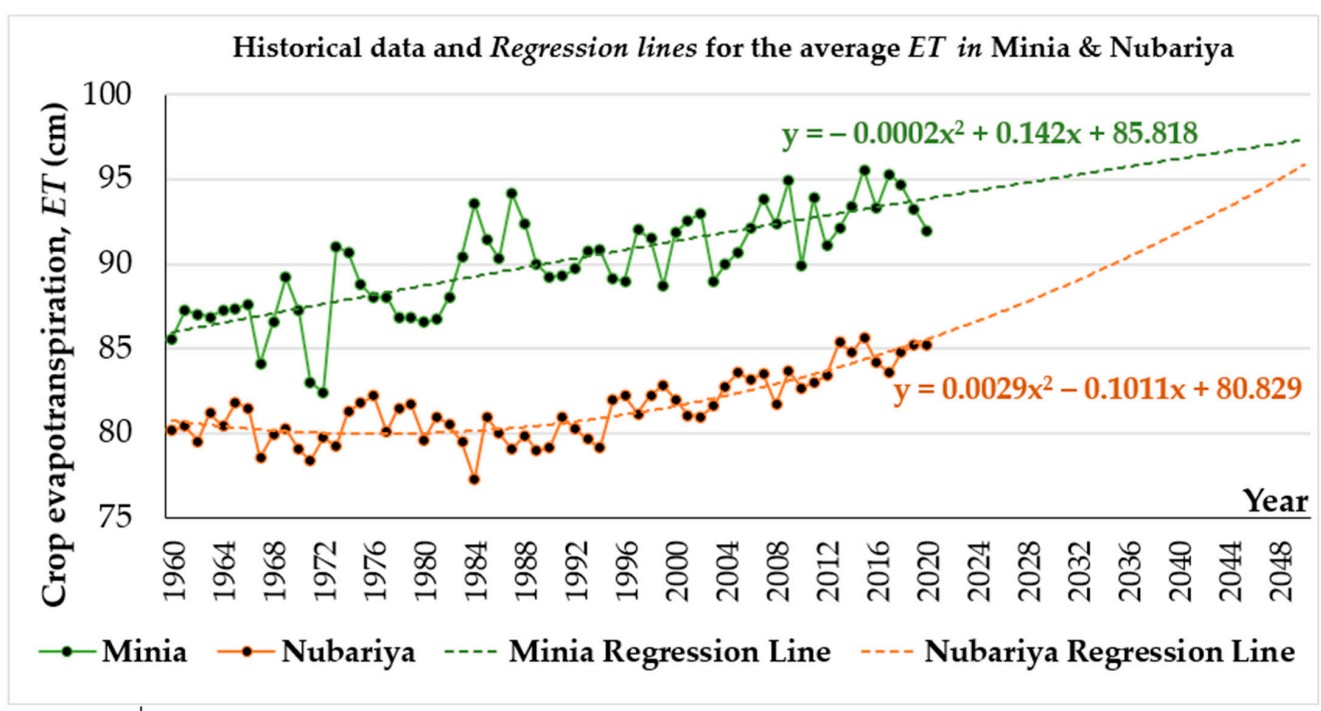

**Figure 12.** Trend analysis of actual ET for historical data from 1960 to 2020, and the linear regression for the predictive values until 2050 for Minia and Nubariya.

**Table 11.** Minimum, maximum, and predictive average temperatures (*T*) and (*ET*) for climate change scenarios in the study areas.

| Variable | Minia | | | Nubariya | | |
|---|---|---|---|---|---|---|
| | I–1: Past (1960–1990) | I–2: Current (1991–2020) | I–3: Predictive (2021–2050) | I–1: Past (1960–1990) | I–2: Current (1991–2020) | I–3: Predictive (2021–2050) |
| *T* Max. (°C) | 37.78 | 36.67 | | 31.67 | 32.78 | |
| *T* Min. (°C) | 5.56 | 7.22 | | 8.33 | 8.33 | |
| *T* Average (°C) | 21.64 | 22.38 | 23.98 | 20.39 | 22.40 | 23.90 |
| *ET* Max. (cm) | 94 | 96 | | 82 | 86 | |
| *ET* Min. (cm) | 82 | 89 | | 77 | 79 | |
| *ET* Avg. (cm) | 89 | 92 | 95.55 | 80 | 83 | 90.414 |

According to Table 11, the average temperature will rise by 1.6 and 1.5 °C in Minia and Nubariya, respectively. The temperature increase values obtained were identical to those in the sixth report of the Intergovernmental Panel on Climate Change (IPCC, 2022); [66,83], and this, in turn, will lead to an increase in *ET* of 5.42% and 5.13% in Minia and Nubariya, respectively. The predictive values of *PE* and *ET* are presented in Tables S6 and S7 in the Supplementary Materials.

### 4.3. Canals Lining Scenarios

According to the two canals that have been observed in the two study areas, the most famous equations were used for estimating the seepage water, and the saved amounts of water from lining are summarized in the following constructed table (Table 12), where (*i*) bed slope (m/m), (*a*) area of wetted perimeter, and (*H*) water depth (m).

The total canal length is 3937.104 km in Minia and 732.52 km in Nubariya. As such, for lining 10% and 25% of the canal lengths in the two case studies, considering 70% seepage reduction, the calculated saved average discharge was found to be:

1.  By lining 10% of canal lengths, the average saved discharge will be 556.24 and 177.88 million cubic meter per year in Minia and Nubariya, respectively.
2.  By lining 25% of canal lengths, the average saved discharge will be 1390.60 and 444.695 million cubic meters per year in Minia and Nubariya, respectively.

**Table 12.** The well-known seepage equations and their affecting parameters on the seepage values.

| No | Equation | Notes | Seepage (Minia) ($m^3/s/Km$) | Seepage (Nubariya) ($m^3/s/Km$) |
|---|---|---|---|---|
| 1 | $S = C_1 LP\sqrt{R}$ [84] | length in km, $C_1$ = 0.003, 0.0015, 0.0018, 0.0022, and 0.0026 for sandy loam, clay, silty clay, clay loam, and silty loam, respect. | 0.053 | 0.103 |
| 2 | $S = \frac{C_2 \times 10^{-4}}{R^{1.166} \times i^{0.5}}$ [78] | $C_2$ = 0.375 and 0.75 for clayey soil and sandy soil, respectively. | 0.09 | 0.18 |
| 3 | $S = 0.55 \times 10^{-6} C_5 PL\sqrt{H}$ [85] | $C_5$ varies between 1.5 and 5.5. | 0.074 | 0.09 |
| 4 | $S = C_6 aH$ [86,87] | $a$ is in million ft2, $C_6$ varies from 1.1 to 1.8 | 0.04 | 0.07 |
| Average | | | 0.064 | 0.11 |

These findings led to the fact that, in Minia, the unmet demand will be zero if 10% of canal lengths are rehabilitated, if the branches of Bahr Youssef Canal are rehabilitated. On the contrary, the unmet demand will be zero if 25% of canal lengths are rehabilitated if the branches of Ayser Al–Banger canal are rehabilitated in the Nubariya region.

It is worth mentioning that the water–food (agriculture)–ecosystem is a fundamental framework to optimize the use of natural resources to meet the growing demand for food [88]. Therefore, a dedicated series of papers focusing on this matter started with an analysis of the detection of groundwater quality in both Minia and Nubariya case studies in Egypt [28], which is an indicator of ecosystem health. While the current study focuses on the developing scenarios of available water use to meet agriculture/food–water demand.

## 5. Conclusions and Recommendations

The calibration of models is important to render their results closer to the observed values in modeling processes. In this research, after we selected two control points beside the main inflow gauges in the two study areas, we entered their discharge values. During the calibration process, the model focused on reducing the difference between the simulated and observed discharges at the gauge stations. After calibrating the model by entering the values for the first site only, then for two sites, and finally for the three sites of observed discharges, it was observed that the simulated values by the models match very well the observed values. At the last run of the model and after choosing the best case among the proposed cases for calibration procedures, the values of the parameters for the canals in the study areas became almost as high as the actual canals flow, which were completely acceptable. After that, the models' validations were performed by comparing the obtained values for known gauge stations in the two study areas from the models with their observed values, which have been obtained from MWRI. The parameters for calibrations and validations, *PBIAS*, *NSE*, *d*, and *VE*, had been calculated for whole gauges in the two study areas and compared with their performance ratings since the results showed "very good" agreement for the models results in terms of accuracy. According to the results, it can be concluded that the third case for calibration in the two models provides a value approach to reality where the validation of the selected sites surface flow shows the models are providing fairly good results. Then, two proposed scenarios were built: climate change and canal lining. The climate change scenario was proposed by tracking past and current temperatures from 1960 to 1990 and from 1991 to 2020 to obtain a prediction of increasing temperatures by 1.6 and 1.5 in Minia and Nubariya, respectively. The finding, in turn, will

lead to an increase in *ET* of 5.42% and 5.13% in the two areas. Canal lining scenarios were proposed for between 10 and 25% of canal length rehabilitations in the two study areas. From the study results, we can conclude the following outcomes:

- It could be stated from the presented results and their study that the models are accurately capable of predicting water demand in the study areas with accuracy up to 95.5% in Minia and up to 97% in Nubariya.
- It was found that the consumptive use of crops will increase water shortages in the two case studies by different ratios depending on the predicted increase in temperatures (5.42% and 5.13%).
- A water shortage could be overcome by rehabilitating 10% of Bahr Youssef canal branches in Minia and 25% of Ayser Al-Banger canal branches in Nubariya.
- It is recommended to test all possible scenarios, including the following scenarios:
  1. Increasing crop consumptive use.
  2. Increasing reuse from drainage water by 7.4%, according to Egypt's National Water Resources Plan (NWRP).
  3. Increasing head flow (in flood seasons).
  4. Lining by 50% of canals served area.
  5. Serving the study areas by modern irrigation (sprinklers and drippers) by different ratios (5%, 10%, and 50% in Minia, and 100% in Nubariya according to Egypt NWRP).
  6. Combination between the previous scenarios to predict best and worst scenarios.

**Supplementary Materials:** The following supporting information can be downloaded at: https://www.mdpi.com/article/10.3390/w15142668/s1, Figure S1: Case 1; Calibrated flows obtained from WEAP Model in Minia; Table S1: Minia Case 1 Calibrated flows obtained from WEAP Model, in which data entry for Ibrahemia inflow only; Figure S2: Case 2; Calibrated flows obtained from WEAP Model in Minia; Table S2: Minia Case 2 Calibrated flows obtained from WEAP Model, in which data entry for Ibrahemia and Maghagha Reservation inflows; Figure S3: Case 1; Calibrated flows obtained from WEAP Model in Nubariya; Table S3: Nubariya Case 1 Calibrated flows obtained from WEAP Model, in which data entry for St.3 inflow only; Figure S4: Case2; Calibrated flows obtained from WEAP Model in Nubariya; Table S4: Nubariya Case 2 Calibrated flows obtained from WEAP Model, in which data entry for St.3 and St.4 inflows; Table S5: Validated flows obtained from WEAP Models in Minia and Nubariya; Table S6: Predictive ET and PE values in Minia from increasing temperature by 1.6 °C; Table S7: Predictive ET and PE values in Nubariya from increasing temperature by 1.5 °C [89].

**Author Contributions:** Conceptualization, A.N., F.A., E.M.R. and A.M.B.; methodology, E.M.R. and A.M.B.; software, E.M.R. and A.M.B.; validation, A.N., E.M.R. and F.A.; formal analysis, A.M.B.; investigation, A.N.; resources, A.M.B.; data curation, E.M.R.; writing—original draft preparation, A.M.B.; writing—review and editing, A.M.B.; visualization, E.M.R. and F.A.; supervision, A.N., E.M.R. and F.A.; project administration, A.N. and F.A.; funding acquisition, F.A. This research is part of the PhD of the first author under the supervision of the coauthors. All authors have read and agreed to the published version of the manuscript.

**Funding:** This study was funded by the EU H2020 PRIMA project SIGMA Nexus—Sustainable Innovation and Governance in the Mediterranean Area for the WEF Nexus, grant #1943 (SIGMANEXUS (Call 2019 Section 1 Nexus RIA)). The funding agency had no role in the study design, data collection, analysis, decision to publish, or preparation of the manuscript. The findings and conclusions in this publication are those of the authors and should not be construed to represent any official EU or PRIMA determination or policy.

**Data Availability Statement:** Not applicable.

**Conflicts of Interest:** The authors declare no conflict of interest.

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
