# Peer review of "Mitigating Water Shortage via Hydrological Modeling in Old and New Cultivated Lands West of the Nile in Egypt"

_water, doi:10.3390/w15142668_

Round 1
Reviewer 1 Report (New Reviewer)
The manuscript is not very bad from a scientific point of view. However, the English and the presentation of the manuscript are very poor. Even it is evident from the title of the article. If the Editor wants to give the authors a chance and if the authors can remarkably improve the English standards, I will be happy to re-review. I have marked my comments/observations in the pdf attached here.

The English standards are very poor. The authors must get the manuscript checked by a native English speaker. At many places, it is even difficult to interpret what exactly the authors are trying to convey.
Author Response
The authors wish to express their gratitude and sincere thanks and regards to the reviewers for their valuable comments that really improved the presentation and quality of the article.
(The modifications were highlighted by yellow colors in the manuscript)
|
List of responses for Reviewer 1 |
|
|
Comments |
Responses |
|
1. ***The title is very poorly written. Just mention the names of the regions; that's enough. *** |
Deep thanks for the respective reviewer. The title was modified from “Mitigating Water Shortage Via Hydrological Modeling in Minia and Nubariya, West of; Nile and Nile Delta, Egypt” to “Mitigating Water Shortage Via Hydrological Modeling in Old and New Cultivated Lands, West of Nile, Egypt”. |
|
2. Middle East, and North Africa (MENA) ***No need to darken the first letters or the abbreviation. There are numerous instances of such issues throughout the manuscript. Please correct them. *** |
Done and all instances of such issues throughout the manuscript had been corrected. |
|
3. Overall, the abstract is too long. This should be reduced. |
Deep thanks for the respective reviewer comment. the abstract has been modified and reduced. |
|
4. No need to quote. This can be easily paraphrased. |
Thanks for drawing attention, the quotations have been removed. |
|
5. Lines 71-110 can be shortened to 1 paragraph (limited to 5-6 sentences). There is no need to describe HYDRUS, SWMM models, or vague descriptions of horological modeling. |
Done. |
|
6. If you are citing a finding, then only numbers can be put at the end of the sentence; however, if you are citing them as a noun in your sentence, you should write the name and then put numbers in brackets. For example, "Psomas et al. [10] combined..." Similar corrections should be done throughout the manuscript. |
Citations were corrected in the whole manuscript |
|
7. The authors described that the WEAP is combined with different models. Why the authors didn't use combination? What is the point of such introduction then? |
The author used WEAP in combination with GIS and mathematical modeling and were clarified through lines 98 – 101 in the manuscript. |
|
8. WEAP, that will assist scenarios and planning evaluation techniques internationally and in areas … “Very poorly written. Please rewrite with clarity.” |
The paragraph has been rewritten, lines 101 – 105. |
|
9. "Nature" is vague. It should be "Land use and soil". |
Thanks for drawing attention, the writing was modified. |
|
10. Please put the full form when the abbreviation is used for the first time. |
Done, line 132. |
|
11. Please improve the resolution of figure 2. |
The figure resolution was improved. |
|
12. Poster … ??? |
Poster was referred to domestic use and were clarified. |
|
So, we obtained their total values… “It is really difficult to understand what the authors are trying to convey. The manuscript needs an extensive editing”. |
The paragraph had been rewritten, lines 197 – 199. |
|
Drinking water stations, ??? |
Drinking water stations was referred to domestic use and were clarified. |
|
In Minia, the main head flow is the Ibrahemia canal inflow… I am astonished to see that the authors are trying to disturb the flow of the manuscript so abruptly.
Please shift this to page 8-9. |
It was done, lines 156 – 157. |
|
Just fed by the main discharge…. Very poor and casual writing. This is not acceptable! |
It was re written in lines 305 – 614. |
|
13. By analyzing the Oshroba canal in Minia…, I don't understand anything from this. The authors must get the manuscript checked by a native English speaker. |
Done in lines 495 – 500. |

Reviewer 2 Report (New Reviewer)
The paper is mainly concerned with irrigation water management in the Nile Valley, and improving the modelling to allow better administration. The paper uses a large suite of methodologies to examine how the water is modelled and distributed.
In many ways the paper is an admirable account of this work and will be of interest to many readers. At the same time it is complex paper and it is hard to follow the many steps in the process, and to judge how well the work is done. I have read and reread the paper and am still confused as to how all the "bits" fit together.
The paper does present a lot of information that will be new to most readers of your journal, and from that point of view is worthy of presentation. It is probably better viewed as a presentation of applied science rather than as a scientific paper in its own right. For that reason I recommend publication with further change (other than some minor editing of English).
Generally a well written paper with few issues here. The authors are to be congratulated. Some of the illustrations are complex and may not necessarily convey their meaning to the readers.
Author Response
The authors wish to express their gratitude and sincere thanks and regards to the reviewers for their valuable comments that really improved the presentation and quality of the article.
(The modifications were highlighted by yellow colors in the manuscript)
|
|
|
Comments |
Responses |
|
1. The paper is mainly concerned with irrigation water management in the Nile Valley and improving the modelling to allow better administration. The paper uses a large suite of methodologies to examine how the water is modelled and distributed. |
Sincere thanks to the respected reviewer. |
|
2. In many ways the paper is an admirable account of this work and will be of interest to many readers. At the same time, it is complex paper, and it is hard to follow the many steps in the process, and to judge how well the work is done. I have read and reread the paper and am still confused as to how all the "bits" fit together. |
The manuscript had been rearranged and the full paper arrangement was modified after all authors responses on the reviewers’ comments. A lot of paragraphs were shortened and clarified, and figures and tables were modified. |
|
3. The paper does present a lot of information that will be new to most readers of your journal, and from that point of view is worthy of presentation. It is probably better viewed as a presentation of applied science rather than as a scientific paper in its own right. For that reason I recommend publication with further change (other than some minor editing of English). |
Water uses in Egypt have large Varity in the hydrological conditions, especially there were old lands (i.e., Minia) cultivated for more than 5000 years, and new cultivated desert area (i.e., Nubariya) similar to many areas all over the world, especially around rivers. the authors tried to propose a feasible mitigation solution to the problem of water scarcity in two different study areas with different hydrological conditions using the same methodology in different locations respect to the Nile River which like a lot of areas all over the world using different mathematical calculation, data collections, and numerical modeling. Then predict the effects of climate change effects on the two different study areas and try to mitigate these effects by using lining of canals. Furthermore, English language was modified. |
|
4. Generally, a well written paper with few issues here. The authors are to be congratulated. Some of the illustrations are complex and may not necessarily convey their meaning to the readers. |
The whole manuscript was modified, and more discussions were clarified. |

Reviewer 3 Report (New Reviewer)
The abstract is too long and descriptive. There is little information about the results in the values. Maybe consider reducing the general descriptions in Lines 14-31.
Introduction: More attention should focus on the research gap in previous studies, which has not been clearly presented in the current introduction. Also, the novelty is not clearly presented.
Double check the required Table format. Keep them consistent throughout the manuscript.
Some figures have poor visibility, and the font sizes in the illustrations are different. Also, the format is not uniform. The authors should improve the resolution of the figures.
How to consider the coupled surface and subsurface hydrological processes? Lines 96-101, a newly developed approach is presented to couple HYDRUS-1D and K2 and to solve the surface-subsurface interaction. In this regard, the author could look at the following reference, “A computationally efficient hydrologic modeling framework to simulate surface-subsurface hydrological processes at the hillslope scale”, in which the coupled processes of overland water flow and infiltration are considered.
Moderate editing of English language required.
Author Response
The authors wish to express their gratitude and sincere thanks and regards to the reviewers for their valuable comments that really improved the presentation and quality of the article.
(The modifications were highlighted by yellow colors in the manuscript)
|
|
|
Comments |
Responses |
|
1. The paper is mainly concerned with irrigation water management in the Nile Valley and improving the modelling to allow better administration. The paper uses a large suite of methodologies to examine how the water is modelled and distributed. |
Sincere thanks to the respected reviewer. |
|
2. In many ways the paper is an admirable account of this work and will be of interest to many readers. At the same time, it is complex paper, and it is hard to follow the many steps in the process, and to judge how well the work is done. I have read and reread the paper and am still confused as to how all the "bits" fit together. |
The manuscript had been rearranged and the full paper arrangement was modified after all authors responses on the reviewers’ comments. A lot of paragraphs were shortened and clarified, and figures and tables were modified. |
|
3. The paper does present a lot of information that will be new to most readers of your journal, and from that point of view is worthy of presentation. It is probably better viewed as a presentation of applied science rather than as a scientific paper in its own right. For that reason I recommend publication with further change (other than some minor editing of English). |
Water uses in Egypt have large Varity in the hydrological conditions, especially there were old lands (i.e., Minia) cultivated for more than 5000 years, and new cultivated desert area (i.e., Nubariya) similar to many areas all over the world, especially around rivers. the authors tried to propose a feasible mitigation solution to the problem of water scarcity in two different study areas with different hydrological conditions using the same methodology in different locations respect to the Nile River which like a lot of areas all over the world using different mathematical calculation, data collections, and numerical modeling. Then predict the effects of climate change effects on the two different study areas and try to mitigate these effects by using lining of canals. Furthermore, English language was modified. |
|
4. Generally, a well written paper with few issues here. The authors are to be congratulated. Some of the illustrations are complex and may not necessarily convey their meaning to the readers. |
The whole manuscript was modified, and more discussions were clarified. |

Reviewer 4 Report (New Reviewer)
1. What is the main innovation in this study? After reading abstract section it is difficulty to find any new contributions. ​2. The intriduction section is too short and it has potential to be extended. ​3. The quality of Figure 1 can be improved and more details can be added. ​4. The flow chart of the methodology can be further revised and reduced. ​5. The result can be re-organised and discussion can be added.Author Response
The authors wish to express their gratitude and sincere thanks and regards to the reviewers for their valuable comments that really improved the presentation and quality of the article.
(The modifications were highlighted by yellow colors in the manuscript)
|
|
|
Comments |
Responses |
|
1. What is the main innovation in this study? After reading abstract section it is difficult to find any new contributions.​ |
Sincere thanks to the respected reviewer. Water uses in Egypt have large Varity in the hydrological conditions, especially there were old lands (i.e., Minia) cultivated for more than 5000 years, and new cultivated desert area (i.e., Nubariya) similar to many areas all over the world, especially around rivers. The authors tried to propose a feasible mitigation solution to the problem of water scarcity in two different study areas with different hydrological conditions using the same methodology in different locations respect to the Nile River which similar to a lot of areas all over the world using different mathematical calculation, data collections, and numerical modeling. Then predict the effects of climate change effects on the two different study areas and try to mitigate these effects by using lining of canals. This was not done before this study. Therefore, it is a unique research work in the field of water shortage mitigation for the two case studies. It was clarified in lines 98 – 105 to cover the research gap in previous studies. |
|
2. The introduction section is too short, and it has potential to be extended.​ |
It was modified. |
|
3. The quality of Figure 1 can be improved and more details can be added.​ |
Done. The figure resolution was increased to 400 dpi |
|
​4. The flow chart of the methodology can be further revised and reduced |
Done |
|
5. The result can be re-organized and discussion can be added. |
The discussion had been modified and enriched. |

Round 2
Reviewer 1 Report (New Reviewer)
The revised manuscript can be accepted for publication. No more comments.
This manuscript is a resubmission of an earlier submission. The following is a list of the peer review reports and author responses from that submission.
Round 1
Reviewer 1 Report
Abir M. Badr et al. conducted a study on mitigating water scarcity in Minia and Nubariya, which are located in the west of the Nile and Nile Delta regions in Egypt, using hydrological modeling. The paper is engaging and well-presented, and it is recommended for publication with some minor modifications. The following are suggestions for the authors:
Major suggestions:
(1) The authors should provide a more in-depth discussion of the study's impact, comparing the WEAP model to other popular models, such as Storm Water Management Model (SWMM) and HYDRUS, and highlighting their respective advantages and disadvantages.
(2) The abstract could be slightly longer, while still maintaining the main message. The authors should consider shortening it.
(3) The presentation could be improved by adding more information to Figure 1 regarding the data collection process. Specifically, the authors should indicate which data are required and the quantity of data needed.
(4) Some part of writing is highly similar to the previous study Elsayed M. Ramadan et al. "Detection of Groundwater Quality Changes in Minia Governorate, West Nile River" (2023) Sustainability 2023, 15(5), 4076
For example, this study Line 17 presents "In the last few years some countries in this region have faced a crisis of water and there is an expectation that the water situation will decline from bad to worse"
Elsayed M. Ramadan et al (2023) presented "In the last few years, some countries in this region have faced a crisis of water and there is an expectancy that the water situation will deteriorate from bad to worse"
The authors may perform a check before publication.
Minor suggestions:
(1) In line 2, the location name "Minia and Nubariya; West of; Nile and Nile Delta; Egypt" should be reviewed as it is not easily located on Google Maps. "Minia (west Nile valley) and Nubariya (west Nile Delta), Egypt" may be more suitable.
(2) In line 7, the first author is identified as "M.Sc.," which is commonly used as a degree title rather than a job title. If the first author has already graduated, their current job title at General Administration of Irrigation West Sharkia should be used.
(3) The font size in Figure 2 (line 174) and Figure 5 (line 259) should be increased for readability.
(4) In Table 4 (line 192), the meaning of "Total" should be clarified as the sum of all locations. The numbers in the table do not add up correctly, so this should be reviewed.
(5) In line 225, "from Dec. to March" should be changed to "from December to March."
The English writing is good.
Author Response
|
General Comments: |
|
|
The authors should provide a more in-depth discussion of the study's impact, comparing the WEAP model to other popular models, such as Storm Water Management Model (SWMM) and HYDRUS, and highlighting their respective advantages and disadvantages. |
Deep thanks for the respective reviewer. We believe that the article has become more organized after considering the comments, modifications and improvements proposed by the reviewers More discussion was provided in Table 1 and Lines 82-88 in the manuscript |
|
(2) The abstract could be slightly longer, while still maintaining the main message. The authors should consider shortening it. |
The abstract had been shortened. |
|
(3) The presentation could be improved by adding more information to Figure 1 regarding the data collection process. Specifically, the authors should indicate which data are required and the quantity of data needed. |
Deep thanks for the respective reviewer comment. more information was added to Figure 1. |
|
(4) Some part of writing is highly similar to the previous study Elsayed M. Ramadan et al. "Detection of Groundwater Quality Changes in Minia Governorate, West Nile River" (2023) Sustainability 2023, 15(5), 4076 |
Thanks for drawing attention, the repeated writing was avoided. |
|
Minor suggestions: |
|
|
(1) In line 2, the location name "Minia and Nubariya; West of; Nile and Nile Delta; Egypt" should be reviewed as it is not easily located on Google Maps. "Minia (west Nile valley) and Nubariya (west Nile Delta), Egypt" may be more suitable. |
The location names were modified, lines 24-26 in the manuscript. |
|
(2) In line 7, the first author is identified as "M.Sc.," which is commonly used as a degree title rather than a job title. If the first author has already graduated, their current job title at General Administration of Irrigation West Sharkia should be used. |
Author job title was modified and added in the manuscript |
|
(3) The font size in Figure 2 (line 174) and Figure 5 (line 259) should be increased for readability. |
All figures were changed to higher quality and font size has been increased. |
|
(4) In Table 4 (line 192), the meaning of "Total" should be clarified as the sum of all locations. The numbers in the table do not add up correctly, so this should be reviewed. |
The word “Total” had been replaced by the word “Sum.). And the numbers were corrected, Page 7, Table 5. |
|
(5) In line 225, "from Dec. to March" should be changed to "from December to March." |
Thanks for drawing attention, the writing were modified, Page 7, Line 197 and Page 8, Line 222. |

Reviewer 2 Report
General comments:
This paper describes an integrated executive system for managing water resources in two different regions in Egypt that have characteristics similar to many areas in MENA region. The simulation of Minia Governorate, western bank of narrow valley of the Nile and Nubariya; West Nile Delta, lower Nile reach, Egypt, network systems were performed using Water Evaluation And Planning (WEAP) model. Satellite interpreted data for digitizing streams on GIS software and various other inputs based on field data and experience. Two scenarios were examined for predicting water shortage through 2020-2050. However, several issues need to be clarified in the revised version.
Specific comments:
1. While there are many data are presented in many tables and figures, they are not systematically organized and summarized in results section, rather they are presented in haphazard manner. In different sub sections under the results section, first results need to be summarized in text followed by data in tables or figures. Also, there are texts in results section which should have been presented in Materials and Methods section. Materials and methods section also has organization and clarity problems. There is quite a bit of information in materials and methods section which should have been included in Introduction section.
2. There are more than recommended numbers of tables and figures. Maximum of 12 tables and/or figures can be included in main text and the remaining moved to supplementary materials file. Quality of tables and figures need to be improved substantially.
3. Introduction requires review of global literature and linking them with Egypt to justify the problem and formulate hypothesis and objectives. Discussion section requires comparing and discussing the results with broader global literature and not just from local area.
4. The innovation of the paper is not clearly stated. The discussion and analysis of the research results are relatively simple. The simulation method and prediction method used in this paper are used more, there is no technical innovation. In case study, the results of the study are not combined with the reality to explain, lack of convincing.
5. The paper should be well organized, and English must be improved. The paper cannot be read smoothly with some awkward sentences. I suggest that the author can employ a professional language editor to improve expression.
6. Citations have to be updated to some newest literature (especially literature from high quality journals must be included). Please note that all the references you mentioned in the text must be listed in the reference list and vice versa.
7. The limitation of the model developed should be described more.
8. The author chooses a single WEAP model for example analysis. Then How to prove the superiority of this model?
9. The manuscript is lengthy and unbalanced. Probably the Introduction and Related work could be shortened a bit and discussion could be enriched with some more insight.
10. Restructuring of the sections is recommended to secure a more logical flow that can facilitate easier reading of the manuscript.
11. The Discussion section is poorly written. At present a lot of it is the replication of the results and it is not enough to fulfill reader's expectation concerning this manuscript. Further proof your results with respect to other studies known for study area and highlight your results in the frame of global and regional climate change.
I recommend that this paper be published with revisions indicated above.
Author Response
|
General Comments: |
|
|
This paper describes an integrated executive system for managing water resources in two different regions in Egypt that have characteristics similar to many areas in MENA region. The simulation of Minia Governorate, western bank of narrow valley of the Nile and Nubariya; West Nile Delta, lower Nile reach, Egypt, network systems were performed using Water Evaluation and Planning (WEAP) model. Satellite interpreted data for digitizing streams on GIS software and various other inputs based on field data and experience. Two scenarios were examined for predicting water shortage through 2020-2050. However, several issues need to be clarified in the revised version. |
Sincere thanks to the respected reviewer. A lot of paragraphs and figures and tables were modified in the cyan color in the manuscript, which help in increasing the clarification of the revised manuscript. |
|
Specific comments: |
|
|
1. (a) While there are many data are presented in many tables and figures, they are not systematically organized and summarized in results section, rather they are presented in haphazard manner. (b) In different sub sections under the results section, first results need to be summarized in text followed by data in tables or figures. (c) Also, there are texts in results section which should have been presented in Materials and Methods section. (d) Materials and methods section also has organization and clarity problems. (e) There is quite a bit of information in materials and methods section which should have been included in Introduction section. |
1.a; The manuscript had been rearranged and the results section was modified. A lot of paragraphs and figures and tables were modified and replaced in the cyan color in the manuscript, Lines 530-537. 1.b; It was rearranged and modified. 1.c; The repeated texts were avoided. 1.d; the Materials section was rearranged. 1.e; A lot of paragraphs were moved to introduction section, Lines 91-128. |
|
2. (a) There are more than recommended numbers of tables and figures. Maximum of 12 tables and/or figures can be included in main text and the remaining moved to supplementary materials file. (b) Quality of tables and figures need to be improved substantially. |
2.a; The authors reduced the numbers of tables and figures. 2.b; All figures were improved to high quality and tables were corrected and some of tables were deleted and their information was added in the text. |
|
3. (a) Introduction requires review of global literature and linking them with Egypt to justify the problem and formulate hypothesis and objectives. (b) Discussion section requires comparing and discussing the results with broader global literature and not just from local area. |
3.a; Global literature and linking them with Egypt to justify the problem were added, Page 4, Lines 130-133 and Page 5, Lines 143-145. 3.b; More global citations were added to the manuscript. |
|
4. (a) The innovation of the paper is not clearly stated. (b) The discussion and analysis of the research results are relatively simple. (c) The simulation method and prediction method used in this paper are used more, there is no technical innovation. (d) in case study, the results of the study are not combined with the reality to explain, (e) lack of convincing. |
4.a; Integration between water distribution models such as WEAP and GIS with mathematical verifications and analytical prediction to climate change. 4.b; The discussion had been modified and enriched, lines 530-539. 4.c; through this research the authors try to achieve the integration between different hydrological study areas with different hydrological situations and parameters. 4.d; The results of the study are combined with the reality in Page 18, Lines 549-550, and Page 19, Lines 573-575. 4.e; the authors tried to propose a feasible mitigation solution to the problem of water scarcity in two different study areas with different hydrological conditions using the same methodology in different locations respect to the Nile River which similar to a lot of areas all over the world using different mathematical calculation, data collections, and numerical modeling. Then predict the effects of climate change effects on the two different study areas and try to mitigate these effects by using lining of canals. |
|
5. (a) The paper should be well organized, and (b) English must be improved. The paper cannot be read smoothly with some awkward sentences. I suggest that the author can employ a professional language editor to improve expression. |
5.a; The paper has been reorganized. 5.b; Revisions for the paper English language were carried out. Also, more clarification for the sentences |
|
6. Citations have to be updated to some newest literature (especially literature from high quality journals must be included). Please note that all the references you mentioned in the text must be listed in the reference list and vice versa. |
literature from high quality journals had been included. And all references were listed in the reference section. |
|
7. The limitation of the model developed should be described more. |
The limitation of the model developed were described in Table 1 |
|
8. The author chooses a single WEAP model for example analysis. Then How to prove the superiority of this model? |
The proposed methodology is represented by coupling GIS and WEAP software with mathematical calculations to represent the past and the current situations and predict the future challenges due to climate and regional changes in two different regions. |
|
9. The manuscript is lengthy and unbalanced. Probably the Introduction and Related work could be shortened a bit and discussion could be enriched with some more insight. |
Manuscript had been rearranged and shortened. |
|
10. Restructuring of the sections is recommended to secure a more logical flow that can facilitate easier reading of the manuscript. |
Manuscript had been modified and rearranged. |
|
11. (a) The Discussion section is poorly written. At present a lot of it is the replication of the results and it is not enough to fulfill reader's expectation concerning this manuscript. (b) Further proof your results with respect to other studies known for study area and highlight your results in the frame of global and regional climate change. |
The discussion section had been rewritten and results of the study are combined with the reality in Page 18, Lines 549-550, and Page 19, Lines 573-575. The results had been modified and enriched, Lines 607-610 |
